# Heme allocation in eukaryotic cells relies on mitochondrial heme export through FLVCR1b to cytosolic GAPDH

Dhanya Thamaraparambil Jayaram [1], Pranav Sivaram[1], Pranjal Biswas[1], Yue Dai [1], Elizabeth A. Sweeny [2] & Dennis J. Stuehr [1] ✉

Heme is an iron-containing cofactor generated in mitochondria that must leave this organelle to reach protein targets in other cell compartments. Because mitochondrial heme binding by cytosolic GAPDH enables its distribution in cells, we sought to uncover how heme reaches GAPDH. Experiments utilizing two human cell lines and a GAPDH reporter protein whose heme binding can be followed by fluorescence reveal that the mitochondrial protein FLVCR1b provides heme to GAPDH in concert with a rise and fall in their association. An absence of FLVCR1b diminishes GAPDH association with mitochondria and prevents GAPDH and cell hemeproteins from receiving heme. GAPDH heme procurement also requires the TANGO2 protein, which interacts with FLVCR1b to presumably support heme export. In isolated mitochondria, GAPDH associates with FLVCR1b to trigger heme release and delivery to client hemeproteins. Identifying FLVCR1b as the source of mitochondrial heme for GAPDH reveals a path by which this essential cofactor can reach multiple protein targets within eukaryotic cells.

Life evolved to utilize a special form of iron bound within the protoporphyrin IX ring (iron protoporphyrin IX; heme)[1,2]. Heme biosynthesis in eukaryotes is a complex process, and its final steps occur inside the mitochondria[3]. How heme is then exported from mitochondria, transported, and inserted into numerous proteins that require heme for function but reside elsewhere in cells has been unclear. Because heme is chemically reactive and has promiscuous binding properties, its synthesis is tightly controlled, and its intracellular transport has long been expected to involve macromolecular carriers[2,4,5]. Of all the proteins or other macromolecules that have been proposed, the protein glyceraldehyde phosphate dehydrogenase (GAPDH), an enzyme in the glycolytic pathway that is ubiquitously expressed and known to perform alternative moonlighting functions[6–8], has recently emerged as a premier intracellular heme chaperone[9], based on findings that GAPDH binding of mitochondrially-generated heme is required for intracellular heme delivery to numerous targets including hemoglobins α, β, and γ[10], myoglobin[10], nitric oxide synthases[11–13] soluble guanylyl cyclase β-subunit (sGCβ)[14], cytochromes P450[15], heme oxygenase 2[16], indoleamine dioxygenase 1 (IDO1) and tryptophan dioxygenase[17]. The insertion of GAPDH-sourced heme into recipient target proteins is the final downstream step in heme delivery, and is now understood to require the cell chaperone protein Hsp90, which is typically bound to the heme-free forms of the recipient proteins and enables their heme insertions in an ATP-driven process[18]. Thus, GAPDH binding of mitochondrial heme allows its wide distribution in cells, most often capped by an Hsp90-assisted heme insertion into the protein target. However, one key question remains unanswered in the overall pathway, namely, how does mitochondrial heme reach GAPDH? Here, we address this question in experiments that utilize two cultured human cell lines, isolated mouse mitochondria, and a newly described GAPDH reporter protein[19] whose heme binding in living cells can be followed in real-time based on the change in its fluorescence intensity. Our results identify the transporter protein Feline Leukemia Virus subgroup C Receptor 1b (FLVCR1b), which we locate to the outer

[1]Department of Inflammation and Immunity, Cleveland Clinic Lerner College of Medicine of Case Western Reserve University School of Medicine, Cleveland, Cleveland, OH, USA. [2]Department of Biochemistry, The Medical College of Wisconsin, Milwaukee, WI, USA. ✉e-mail: stuehrd@ccf.org

mitochondrial membrane, as the major conduit for mitochondrial heme export to GAPDH in the cells. We also show that the protein TANGO2, likely through an association with FLVCR1b in the mitochondria, enables the FLVCR1b heme transfer to GAPDH. Overall, our findings elucidate a major route of heme export from the mitochondria and show how this action is coupled to heme distribution and insertion into final destination proteins in eukaryotic cells.

## Results

### FLVCR1b supplies heme to GAPDH for downstream delivery

FLVCR1b is an integral transmembrane protein that specifically localizes to the mitochondria and enables mitochondrial heme export in erythrocytes during erythropoiesis[20]. To assess its importance in providing mitochondrial heme to GAPDH, we transfected the human cell lines HEK293T and HeLa to express an HA-tagged human GAPDH reporter protein called HA-TC-GAPDH[19], which after being labeled with FlAsH reagent[19,21] signals its heme binding in live cells or in solution by a fluorescence quenching of its FlAsH signal[19]. We then utilized a siRNA that can specifically knock down expression of FLVCR1b but otherwise was reported not to cause off-target effects[20] to determine how FLVCR1b knockdown would impact heme loading into HA-TC-GAPDH in cells upon stimulating their mitochondrial heme biosynthesis. This was accomplished by adding the heme precursor molecules δ-aminolevulinic acid and ferric citrate (δ-ALA/Fe)[22,23], which in our hands typically causes the heme content of HEK293T cells to increase by 3-fold over 2 h and causes a similar increase in the bound heme content of HA-TC-GAPDH[19]. The levels of FLVCR1b protein expression in mitochondrial samples prepared from HEK293T or HeLa cells receiving the targeted siRNA treatment were diminished by 70% ($n = 5$) and 75% ($n = 3$), respectively, relative to mitochondrial samples from cells treated with the scrambled siRNA control (Fig. 1A, Supplementary Figs. S1A, S2A, B). The targeted siRNA treatment did not diminish cell expression levels of HA-TC-GAPDH (Supplementary Figs. S1B, S2C) or the related protein FLVCR1a (Supplementary Fig. S1C, D), consistent with the high specificity reported for this siRNA[20].

HEK293T cells that underwent the FLVCR1b knockdown showed a shift in their intracellular heme distribution such that their mitochondrial heme levels were increased and their cytosol levels were decreased relative to control cells (Supplementary Fig. S1E, F), despite the knockdown causing no change in the cell total heme level (Supplementary Table S1). This is consistent with previous findings by Chiabrando et al.[20] and with FLVCR1b enabling mitochondrial heme export to the cytosol. Indeed, in HEK293T or HeLa cells, the FLVCR1b knockdown inhibited mitochondrial heme transfer to HA-TC-GAPDH by more than 90%. This was indicated by their HA-TC-GAPDH fluorescence emission remaining stable with time after the δ-ALA/Fe addition compared to the decrease in fluorescence emission seen for cells that expressed HA-TC-GAPDH but received either scrambled or no siRNA (Fig. 1B and Supplementary Fig. S2D). Transfecting the HEK293T cells that had undergone the FLVCR1b knockdown with a FLVCR1b expression plasmid successfully rescued their FLVCR1b protein expression (Fig. 1A) and also rescued their mitochondrial heme transfer to HA-TC-GAPDH after the δ-ALA/Fe addition (Fig. 1B). Importantly, knockdown of FLVCR1b expression (Supplementary Fig. S3A–C) did not prevent the HEK293T cells from transferring externally added heme to HA-TC-GAPDH despite inhibiting the transfer of their mitochondrially-generated heme (Fig. 1E, F). Together, these data reveal that FLVCR1b is needed for the HEK293T and HeLa cells to transfer mitochondrial heme to cytosolic HA-TC-GAPDH.

We next examined if the FLVCR1b knockdown would impact downstream delivery of mitochondrial heme to two GAPDH client hemeproteins expressed in the HEK293T cells (a TC-tagged version of soluble guanylyl cyclase β-subunit termed TC-sGCβ, and indoleamine dioxygenase-1, IDO1)[14,17]. The FLVCR1b knockdown did not alter expression of the target hemeproteins in the cells (Supplementary

Fig. S4) but did completely block their heme deliveries following the δ-ALA/Fe addition, as indicated by preventing a decrease in FlAsH-TC-sGCβ fluorescence and preventing an increase in the heme-dependent enzymatic activity of IDO1 (IDO1 conversion of L-Trp to L-kynurenine)[17] (Fig. 1C, D). Thus, FLVCR1b was needed for GAPDH to obtain mitochondrial heme in HEK293T and HeLa cells and was needed for subsequent heme delivery to two heme protein targets in the HEK293T cells, revealing that in the absence of FLVCR1b-supported heme transfer to GAPDH the cells had no alternative means to accomplish these heme deliveries within the timeframe of our experiments.

### FLVCR1b resides in the outer and inner mitochondrial membranes

Given the importance of FLVCR1b in GAPDH-mediated heme allocation, we next investigated its location in mitochondria, where it could conceivably integrate into the outer or inner membranes, or both. We isolated mouse brain and liver mitochondria and performed a proteinase K treatment, which is an established means to identify outer membrane proteins[24], and we also performed a standard mitochondrial sub-fractionation[22,25] followed by Western blotting to locate the FLVCR1b in the inner and outer membrane fractions. The proteinase K treatment eliminated TOM 70, a known outer mitochondrial membrane protein[26], and reduced the mitochondrial FLVCR1b content by 75% while not altering the level of COX4, a protein known to be located exclusively to the inner membrane[27] (Supplementary Fig. S5A, B). This suggested a significant proportion of FLVCR1b was present in the outer mitochondrial membrane. The mitochondrial sub-fractionation procedure confirmed that FLVCR1b was present in the outer membrane along with TOM 70 and additionally revealed that it was present in the inner membrane along with COX4 (Supplementary Fig. S5C, D). A dual membrane location for FLVCR1b would be consistent with its functioning to export heme from the mitochondria and to cytosolic GAPDH.

### FLVCR1b and GAPDH associate during mitochondrial heme export

We next investigated if mitochondrial heme export to GAPDH involved an association with FLVCR1b by utilizing the Duolink® proximity ligation assay (PLA), which can detect interaction between two proteins within a 0–40 nm distance, a typical range for protein-protein associations[28]. We provided δ-ALA/Fe to the HEK293T or HeLa cells to stimulate their mitochondrial heme biosynthesis and then used PLA to assess the level of FLVCR1b-GAPDH association at time points before, during, and after the resultant mitochondrial heme transfer into HA-TC-GAPDH takes place (at 0, 30, 120 min, respectively, see Fig. 1E, F). Supplementary Fig. S6 shows that the background PLA signals generated with Abs directed against FLVCR1b and the noninteracting protein laminin[29] were negligible in the HEK293T or HeLa cells. In comparison, PLA data using antibodies directed against GAPDH and FLVCR1b indicate that a detectable but low level of GAPDH-FLVCR1b association existed in both HEK293T and HeLa cells under resting conditions, that increased 6 to 7-fold by 30 min after the δ-ALA/Fe addition, and then decayed to near original levels by 120 min (Figs. 2A, B, S7A, B). To determine if an association of GAPDH with the related cell membrane protein FLVCR1a[30] was also involved, we ran replica PLA experiments using antibodies against GAPDH and FLVCR1a. No significant increase in their association occurred after the δ-ALA/Fe addition (Supplementary Fig. S8A, B), indicating the prior PLA results only reflected the change in GAPDH association with FLVCR1b. Immunoprecipitations independently confirmed that the GAPDH-FLVCR1b association rose and fell after the δ-ALA/Fe addition (Fig. 2C, D, Supplementary Figs. S9A–C). In HEK293T or HeLa cells that underwent siRNA knockdown of FLVCR1b expression, the PLA signals for GAPDH-FLVCR1b association were diminished and did not increase

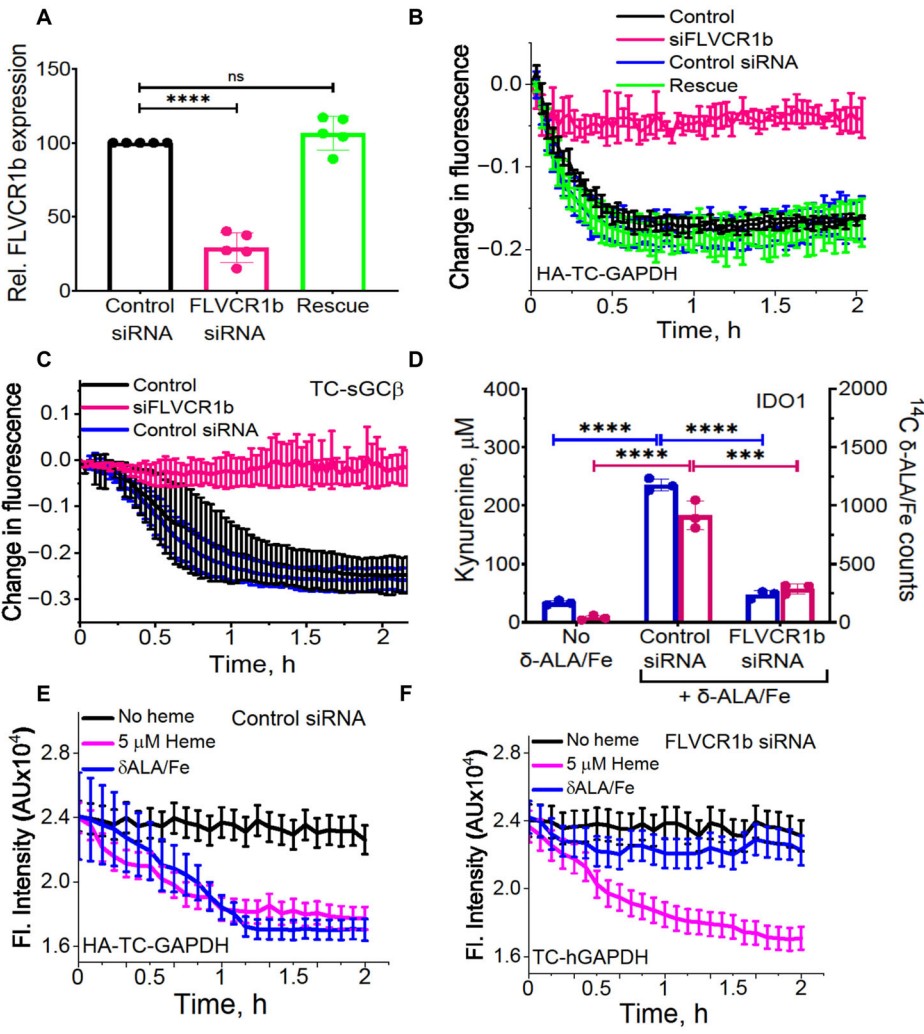

**Fig. 1 | FLVCR1b mediates mitochondrial heme transfer to GAPDH for heme delivery in cells.** HEK293T cells were transfected with scrambled (control) or FLVCR1b-targeted siRNA for 48 h and then were transfected to express FLVCR1b (rescue), HA-TC-GAPDH, TC-sGCβ, or IDO1 as indicated. **A** FLVCR1b expression in mitochondrial samples from the scrambled or FLVCR1b siRNA-treated HEK293T cells without or following rescue of FLVCR1b expression. Samples underwent SDS-PAGE and Western analysis. Mean ± SD of 5 independent trials. Significance: ns ($p = 0.4102$) and **** $p < 0.0001$ vs. the compared group based on a one-way ANOVA test. $F = 121$, DF = 12. **B** Impact of FLVCR1b knockdown and rescue on mitochondrial heme transfer to FlAsH-HA-TC-GAPDH in HEK293T cells after adding δ-ALA/Fe at time = 0. Representative of three trials, mean ± SD of triplicates. Impact of FLVCR1b knockdown on delivery of mitochondrial heme to two heme-proteins after giving δ-ALA/Fe to the HEK293T cells at time = 0, as judged by **C** it

preventing a decrease in FlAsH-TC-sGCβ fluorescence and **D** it blocking an increase in heme-dependent IDO1 enzymatic activity. *p*-values: In Kynurenin assay, Control siRNA + no Heme vs. Control siRNA + Heme $p = <0.0001$, Control siRNA + Heme vs. siFLVCR1b + Heme $p = <0.0001$. In ¹⁴C heme count, Control siRNA + no Heme vs. Control siRNA + Heme $p = <0.0001$, Control siRNA + Heme vs. siFLVCR1b + Heme $p = 0.0001$. **C** Representative of three independent trials, mean ± SD of triplicates. **D** Mean ± SD of 3 independent trials. Significance: *** $p < 0.001$ and **** $p < 0.0001$ vs. the compared group based on a one-way ANOVA test. $F = 109.1$, DF = 6 for ¹⁴C data and $F = 681.9$, DF = 6 for activity of IDO1. **E, F** Cell FlAsH fluorescence was followed versus time (data points are mean ± SD, $n = 3$ wells) after the scrambled or FLVCR1b siRNA-transfected cell cultures received vehicle alone or plus heme (5 μM) or δ-ALA/Fe. Abbreviations: HA-TC Hemagglutinin-Tetra cysteine, δ-ALA/Fe δ-aminolevulinic acid and ferric citrate, μM micromolar.

after the δ-ALA/Fe addition (Fig. 2E, F, Supplementary Figs. S7C, D). However, the gain and fall in GAPDH-FLVCR1b association in response to δ-ALA/Fe addition were restored upon FLVCR1b expression rescue in the knockdown HEK293T cells (Fig. 2G, H). PLA data obtained using antibodies directed against GAPDH and the known mitochondrial surface protein hexokinase-1 showed that FLVCR1b expression was needed for GAPDH to increase its interaction with the mitochondrial surface in response to the δ-ALA/Fe addition in HEK293T or HeLa cells (Supplementary Fig. S10A–D). Thus, GAPDH interaction with the mitochondrial surface in these cells was linked to their expression of FLVCR1b, and FLVCR1b enabled the GAPDH-mitochondria interaction to increase when the cell's mitochondrial heme biosynthesis was stimulated and the heme was being actively transferred into the GAPDH.

We next studied the H53A variant of HA-TC-GAPDH, which has a 30-fold poorer heme binding affinity due to it missing the heme-ligating His53[12], and when expressed in cells it does not accumulate mitochondrially-generated heme in response to δ-ALA/Fe addition[12,19]. After confirming H53A HA-TC-GAPDH has defective intrinsic and intracellular heme binding (Supplementary Fig. S11A, B), we examined its association with FLVCR1b by the PLA method in HEK293T cells. H53A HA-TC-GAPDH displayed some basal association with FLVCR1b but the addition of δ-ALA/Fe to the cells only caused a muted delayed increase in its FLVCR1b association relative to that of wild-type HA-TC-GAPDH, and immunoprecipitation experiments confirmed the PLA results (Supplementary Fig. S11C–F). Thus, a defective mitochondrial association was linked to the poor heme binding affinity of H53A HA-

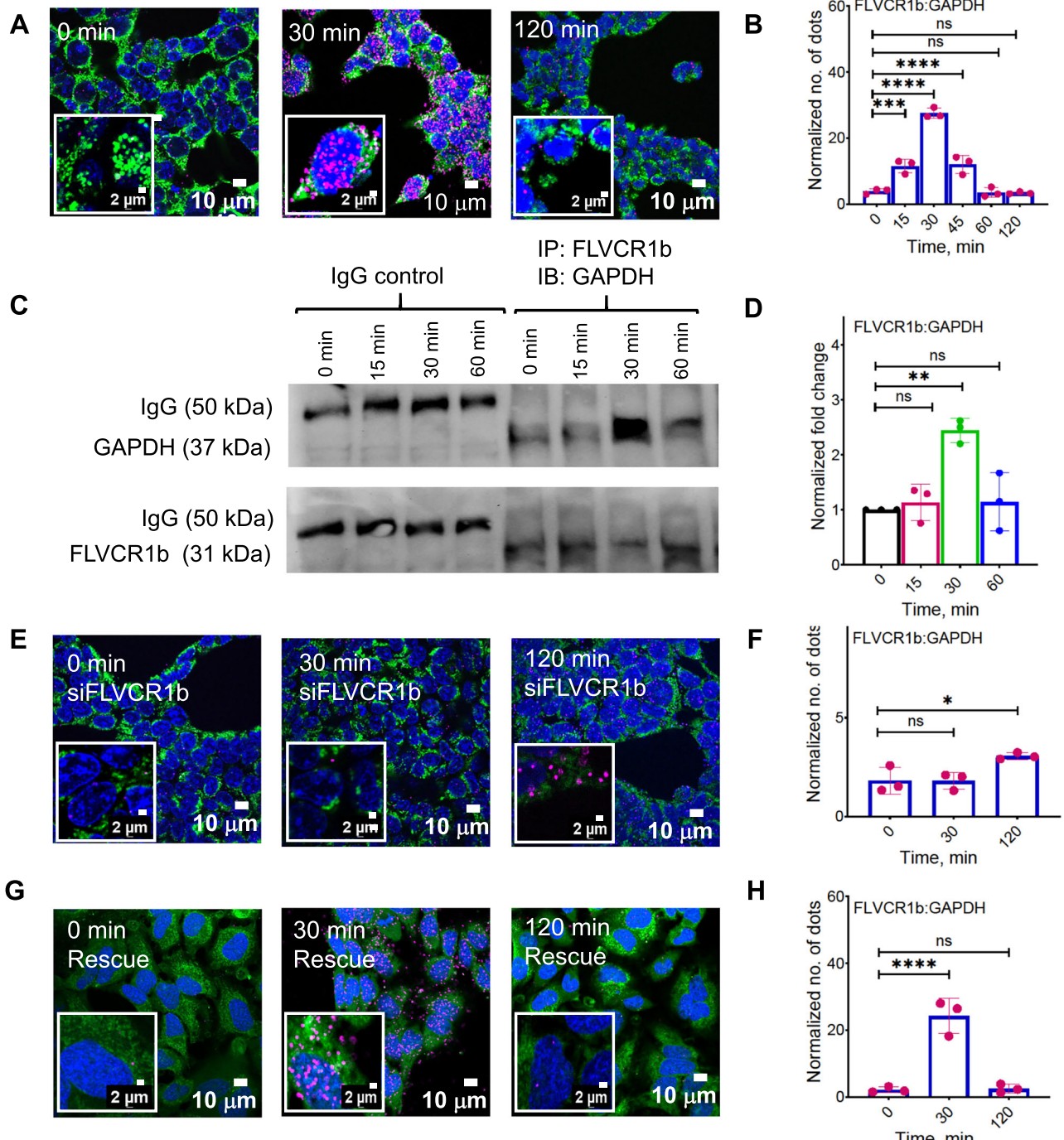

**Fig. 2 | FLVCR1b associates with GAPDH during mitochondrial heme transfer in cells.** Proximity ligation assay (PLA) was used to compare the extent of GAPDH association with FLVCR1b at indicated time points in HEK293T cells after giving them δ-ALA/Fe at time = 0 to stimulate mitochondrial heme biosynthesis. After fixation the cells were stained with DAPI for nuclei (blue), HADHA antibody for mitochondria (green), and GAPDH and FLVCR1 antibodies to show their association by PLA (pink). **A**, **E**, **G** Representative fluorescence microscope images of the control, FLVCR1b knockdown, and FLCVR1b knockdown and FLVCR1b-retransfected rescued cells, respectively. $n = 3$ **B**, **F**, **H** Quantification of PLA signals in the cells under the indicated conditions and times. **C** Immunoprecipitations indicating the time course of FLVCR1b association with GAPDH after stimulation of cell mitochondrial heme biosynthesis. Representative Western blots show relative levels of FLVCR1b and GAPDH proteins in the IP samples (right 4 lanes). Antibody against IgG Isotype control was also used (left 4 lanes), $n = 3$. **D** Fold change in the GAPDH interaction with FLVCR1b versus time after δ-ALA/Fe addition for each IP sample. All microscopic images were quantified using Volocity 6.5.1 (Quorum Technologies). Normalization of PLA signals involved subtracting the background PLA signal obtained using antibodies against FLVCR1b and a non-interacting partner protein (laminin). Statistics; **B**, **D**, **F**, **H** Mean ± SD of 3 independent trials. Significance: * $p < 0.05$, ** $p < 0.01$, *** $p < 0.001$, and **** $p < 0.0001$ vs. the compared group based on a one-way ANOVA test. ns, not significant. $p$-values: **B** 0 vs. 15 min $p = 0.0002$, 0 vs. 30 min $p = <0.0001$, 0 vs. 45 min $p = <0.0001$, 0 vs. 60 min $p = 0.9995$, 0 vs. 120 min $p = 0.9965$. **D** $p$-values: 0 vs. 15 $p = 0.9199$, 0 vs. 30 $p = 0.0017$, 0 vs. 60 $p = 0.899$. **F** $p$-values: 0 vs. 30 min $p = >0.9999$, 0 vs. 120 min $p = 0.0145$. **H** $p$-values 0 vs. 30 min $p = <0.0001$, 0 vs. 120 min $p = 0.9971$. **B**. F = 103.0, DF = 14. **D**. F = 12.83, DF = 8. **F**. F = 12.96, DF = 8. **H**. F = 49.91, DF = 8.

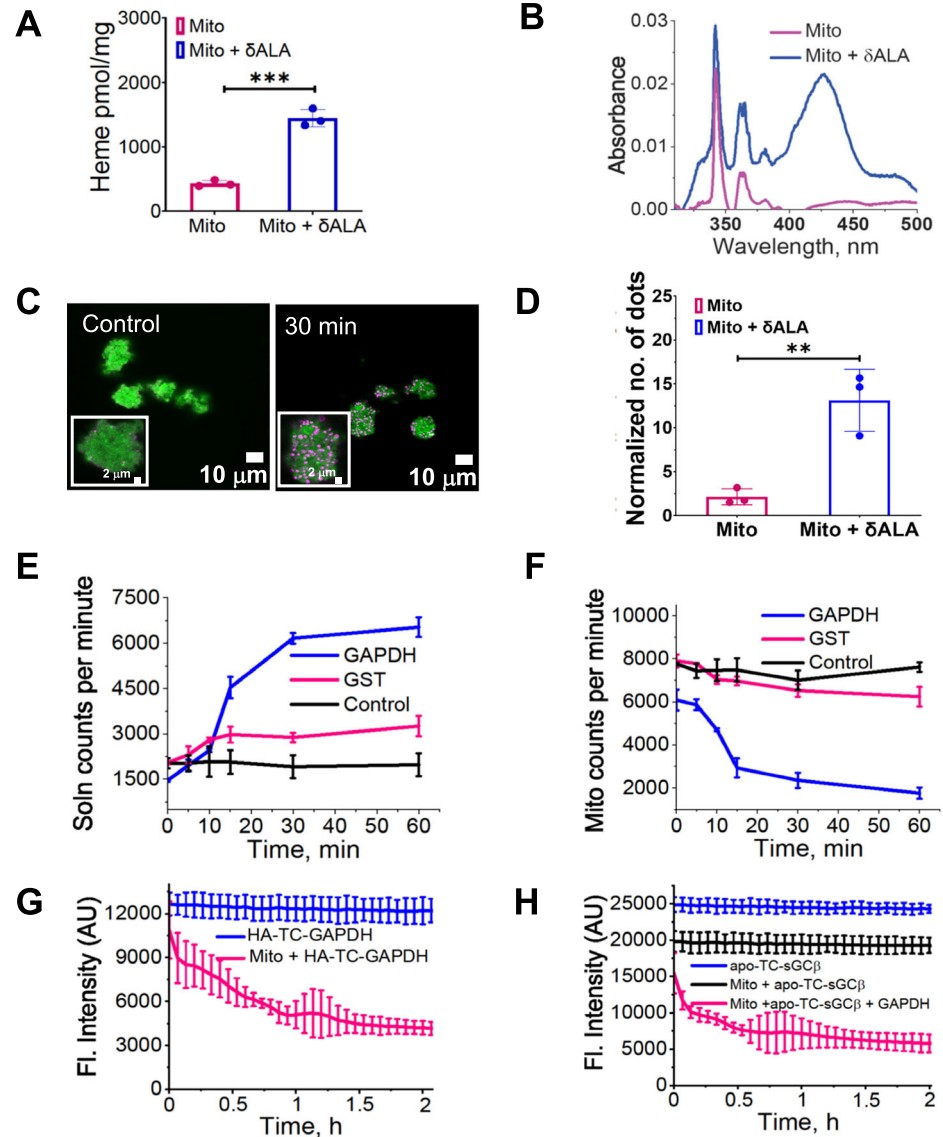

**Fig. 3 | Heme export from isolated mitochondria is stimulated by GAPDH and involves FLVCR1b-GAPDH association and heme transfer to GAPDH.** Mitochondria purified from mouse brain and liver underwent incubation in buffer alone or containing δ-ALA, ADP, and mitochondria-free HEK293T cell supernatant to stimulate their in vitro heme biosynthesis. Heme content of the reisolated mitochondria as indicated by **A** the hemochromogen assay, $t = 12.20$, DF = 4 and **B** by the intensity of the visible spectra 420 nm peak. **C** Representative fluorescence microscope images of mitochondria indicating relative levels of GAPDH-FLVCR1b association (HADHA, green; PLA, pink) before or after the 30 min incubation with δ-ALA, ADP, and cell supernatant followed by reisolation, $n = 3$. **D** Quantification of the PLA results from (**C**) and two additional independent trials, using Volocity 6.5.1 (Quorum Technologies) image analysis software. $t = 5.217$, DF = 4. **E, F** kinetics of [14]C-heme release from mitochondria into solution and loss of mitochondrial [14]C-heme content, respectively, after adding purified GAPDH or GST proteins to reisolated mitochondria that had undergone incubation with [14]C δ-ALA, ADP, and

cell supernatant. **G** Heme transfer to FlAsH-HA-TC-GAPDH upon addition of reisolated mitochondria that had been incubated with δ-ALA, ADP, and cell supernatant. FlAsH-HA-TC-GAPDH in buffer was the control, and fluorescence emission was monitored at 528 nm. **H** Effect of adding GAPDH on heme transfer from reisolated mitochondria to target protein FlAsH-apo-TC-sGCβ. FlAsH-apo-TC-sGCβ in buffer received either nothing (control), reisolated mitochondria that had been incubated with δ-ALA, ADP, and cell supernatant, or these mitochondria plus GAPDH added after five additional min. Fluorescence readings began for all samples after the GAPDH addition (Time = 0). **A, D, E, F**, Mean ± SD of three independent trials. **G, H** Representative of three independent trials, mean ± SD of triplicates. Significance: ** $p < 0.01$, *** $p < 0.001$ vs. the compared group based on a two-tailed $t$ test. $p$-values: **A** Mito vs Mito + δALA $p = 0.0003$. **D** Mito vs Mito + δALA $p = 0.0064$. Abbreviations: Mito Mitochondria, Mito + δALA Mitochondria + δ aminolevulinic acid, HA-TC Hemagglutinin-Tetra cysteine-.

TC-GAPDH and together may likely explain why it was unable to receive mitochondrial heme in the cells.

### GAPDH-FLVCR1b binding on isolated mitochondria triggers heme export and delivery

We next performed experiments with mitochondria freshly isolated from mouse brain and liver to expand on our cell culture findings. Upon incubating the mitochondria for 30 min with mitochondria-free HEK293T cell supernatant plus δ-ALA and their re-isolation, we

observed a 3-fold increase in their heme level (Fig. 3A, B), confirming earlier reports that in vitro mitochondrial heme biosynthesis takes place in this circumstance[31]. PLA experiments done with the re-isolated mitochondria showed that those that underwent the 30 min incubation with δ-ALA and cell supernatant displayed a six-times greater FLVCR1b-GAPDH interaction (Fig. 3C, D), mirroring what we observed in living cells 30 min after stimulating their heme biosynthesis by δ-ALA/Fe addition. In follow-up experiments we incubated isolated mitochondria for 30 min with cytosol and [14]C-δ-ALA so they would

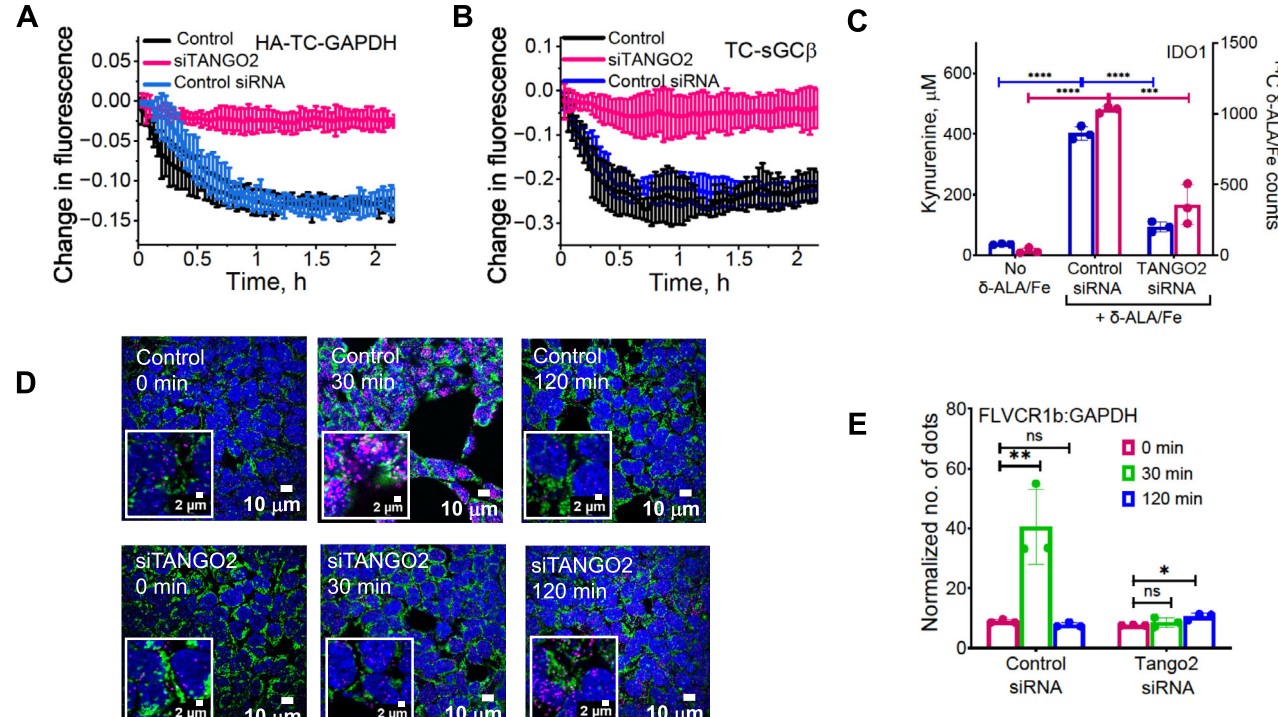

**Fig. 4 | TANGO2 enables FLVCR1b heme export to GAPDH and its delivery to target proteins.** HEK293T cells treated with nothing, scrambled siRNA (control), or TANGO2 siRNA for 48 h were then transfected to express HA-TC-GAPDH or IDO1 as indicated. **A** kinetics of mitochondrial heme transfer to FlAsH-HA-TC-GAPDH in HEK293T cells that had undergone the indicated treatments, after stimulating their heme biosynthesis with δ-ALA/Fe added at time = 0. Effect of TANGO2 knockdown on delivery of mitochondrial heme to two client proteins in cells, as judged by **B** it blocking the fluorescence decrease of FlAsH-TC-sGCβ versus time or **C** blocking the gain in heme-dependent IDO1 activity (L-Kynurenine accumulation, $F = 447.8$, DF = 6) and the gain in IDO1 [14]C-heme content ($F = 110.5$, DF = 6). **D** Fluorescence microscope images of HEK293T cells treated with control or TANGO2 siRNA for 48 h and then incubated with δ-ALA/Fe for the indicated times. Cells stained with DAPI for nuclei (blue), HADHA antibody for mitochondria (green), and antibodies against GAPDH and FLVCR1b to indicate their association by PLA (pink). **E** Quantification of the PLA results from (**D**) and three additional trials, using Volocity 6.5.1 (Quorum Technologies) software. Normalization of PLA signals utilized antibodies against FLVCR1 and laminin. Control cells, $F = 19.84$, DF = 6 and TANGO2-silenced cells, $F = 5.352$, DF = 6. **A**, **B** Representative of three independent trials, values = mean ± SD of triplicates. **C**, **E** Three independent trials, mean ± SD. Significance: * $p < 0.05$, ** $p < 0.01$, *** $p < 0.001$, and **** $p < 0.0001$ vs. the compared group based on a one-way ANOVA test. ns, not significant. *p*-values: **C** In Kynurenin assay, Control siRNA+no Heme vs. Control siRNA+Heme $p = <0.0001$, Control siRNA+Heme vs. siFLVCR1b+Heme $p = <0.000$. In heme count, Control siRNA+no Heme vs. Control siRNA+Heme $p = <0.0001$, Control siRNA+Heme vs. siFLVCR1b+Heme $p = 0.0001$. Abbreviations: HA-TC hemagglutinin-tetra cysteine, δ-ALA/Fe δ-aminolevulinic acid and ferric citrate, μM micromolar.

generate and accumulate radiolabeled [14]C-heme[31], and then re-isolated these mitochondria for experiments. Upon their placement in buffer solution at 37 °C, the mitochondria released an inconsequential amount of [14]C counts into the solution over a 60 min observation period, as indicated by the [14]C counts present in both the solution and the re-isolated mitochondria remaining steady with time (Fig. 3E, F). Adding purified GAPDH at time = 0 resulted in an immediate and time-dependent release of the [14]C counts from the mitochondria into the solution, whereas adding a comparable molar amount of glutathione S-transferase, which had previously been reported to stimulate mitochondrial heme release[32,33] caused much less [14]C counts release (Fig. 3E, F). A hemochromogen test[34] confirmed that the counts released by mitochondria into solution were due to [14]C-heme (Supplementary Table S2).

We then performed an experiment in which purified FlAsH-HA-TC-GAPDH protein was added to the re-isolated heme-loaded mitochondria to determine if the heme release caused by GAPDH addition was tied to it binding to GAPDH. Addition of FlAsH-HA-TC-GAPDH stimulated the release of heme, which bound to it as indicated by the time-dependent decrease in its fluorescence intensity (Fig. 3G). Finally, we examined if adding GAPDH to the reisolated mitochondria would enable heme to be delivered to a GAPDH client protein (FlAsH-TC-apo-sGCβ). As shown in Fig. 3H, the fluorescence intensity of FlAsH-TC-apo-sGCβ in solution remained steady over a 1 h period. When the re-

isolated mitochondria were added it caused an immediate partial decrease in the FlAsH-TC-apo-sGCβ fluorescence which did not change with time and that we attribute to some released heme that had built up in the mitochondrial preparation after their re-isolation. Adding GAPDH to this solution stimulated additional heme to transfer into the FlAsH-TC-apo-sGCβ as indicated by a time-dependent loss of its FlAsH fluorescence intensity (Fig. 3H). Thus, the main features of mitochondrial heme export to GAPDH and subsequent heme delivery to a target protein that we observed in living cells (i.e., a buildup of the FLVCR1b-GAPDH association that coincides with heme transfer to GAPDH and subsequent GAPDH heme transfer to the apo-sGCβ target protein) could be replicated in the simple reconstitution system consisting only of isolated mitochondria, purified GAPDH, and purified apo-sGCβ target protein. In this system, GAPDH associated with mitochondrial FLVCR1b and stimulated release of mitochondrial heme, which bound to the GAPDH and was then transferred to the TC-apo-sGCβ target protein.

**Assessing roles for PGRMC2 and TANGO2 in heme export to GAPDH and allocation**

Because the proteins progesterone receptor membrane component 2 (PGRMC2) and Transport and Golgi Organization 2 (TANGO2) were recently implicated in intracellular heme trafficking[35–38], we tested their involvement in our system. A siRNA knockdown of PGRMC2

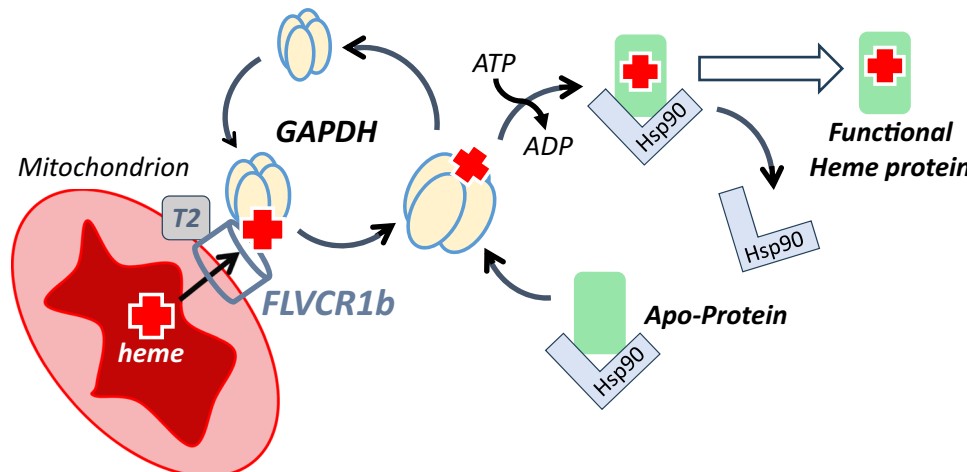

**Fig. 5 | A pathway for mitochondrial heme provision in eukaryotic cells.** Heme made in mitochondria transfers to cytosolic GAPDH through the mitochondrial FLVCR1b exporter, which functions in conjunction with TANGO2 (T2). Heme export involves GAPDH associating with FLVCR1b on the mitochondrial surface to an extent that is influenced by mitochondrial heme biosynthesis and by heme loading into GAPDH. After GAPDH obtains heme, it can dissociate from the mitochondria and make heme available to client hemeproteins located in the cytosol or in other cell compartments. The heme-free clients are typically in complex with cell chaperone Hsp90, which is needed to drive their heme insertions in an ATP-dependent process. Hsp90 then dissociates, enabling the biological function of the hemeproteins.

expression in HEK293T cells reduced PGRMC2 expression by an average of 75% (Supplementary Fig. S12A, B) but did not impact cell expression of other relevant proteins (Supplementary Fig. S12C–E) nor did it inhibit mitochondrial heme transfer to HA-TC-GAPDH upon δ-ALA/Fe addition or inhibit the delivery of mitochondrial heme to downstream target proteins sGCβ or IDO1 in the cells (Supplementary Fig. S13A–C). This indicated PGRMC2 was not involved, and so it was not studied further. In contrast, a siRNA knockdown of TANGO2 expression reduced its level in HEK293T cells by 75% (Supplementary Fig. S14A, B) and strongly inhibited their mitochondrial heme transfer to HA-TC-GAPDH following δ-ALA/Fe addition (Fig. 4A) and their downstream heme deliveries to FlAsH-TC-sGCβ and to IDO1 (Fig. 4B, C) without impacting the expression levels of these proteins (Supplementary Fig. S14C–E) or the final heme level in the cells (Supplementary Table S3). Thus, TANGO2 appeared to also be involved in mitochondrial heme transfer to GAPDH, and we further investigated its role.

Protein association studies using the PLA approach showed that knockdown of TANGO2 expression in HEK293T cells did not greatly alter the level of GAPDH interaction with FLVCR1b in resting cells, but it did prevent their interaction from increasing after δ-ALA/Fe addition (Fig. 4D, E). Further interrogation showed that FLVCR1b associated with TANGO2 in the cells at five times above its background association with Laminin as determined by PLA, and the level of their association remained constant after the cells received δ-ALA/Fe to increase mitochondrial heme production (Supplementary Fig. S15A, B). These results were confirmed by IP (Supplementary Fig. S15C, D). Knockdown of cell GAPDH expression also did not alter the level of TANGO2-FLVCR1b association (Supplementary Fig. S16A–C). PLA results showed that TANGO2 engaged in a weaker association with GAPDH (3.5 times greater than the background interaction between TANGO2 and Laminin) both prior to and following δ-ALA/Fe addition to the HEK293T cells (Supplementary Fig. S17). Together, these results suggested that TANGO2 may associate with FLVCR1b to enable its mitochondrial heme transfer to GAPDH.

Because TANGO2 is known to be present in both the cell cytosol and mitochondria[39,40], we further investigated its role by utilizing isolated mitochondria to examine the importance of cytosolic versus mitochondrial TANGO2 in enabling the mitochondrial heme transfer to HA-TC-GAPDH. We incubated isolated mitochondria with δ-ALA and with mitochondria-free cytosols that were prepared either from normal or from TANGO2 knockdown HEK293T cells, and then reisolated the mitochondria after the incubation and compared the extents of their FLVCR1b-GAPDH interaction by PLA and their abilities to transfer heme to added HA-TC-GAPDH. The two sets of reisolated mitochondria behaved identically in their having increased levels of FLVCR1b-GAPDH interaction (Supplementary Fig. S18A, B) and in their transfer of heme to FlAsH-labeled HA-TC-GAPDH (Supplementary Fig. S18C). These results discounted a role for cytosolic TANGO2 and instead implicate mitochondrial TANGO2 in enabling FLVCR1b heme transfer to GAPDH.

## Discussion

How heme produced inside the mitochondria is released, distributed, and inserted into proteins that reside outside this organelle but need heme to function has been a long-standing question in cell biology[1,4,5,41]. Our finding that the integral membrane protein FLVCR1b is present in both the outer and inner mitochondrial membranes and is needed for heme to reach GAPDH in the cytosol helps to clarify how heme is allocated in mammals. Overall, the current and previous results support a mechanism outlined in Fig. 5: After heme is synthesized inside the mitochondria, it is made available to cytosolic GAPDH through the actions of the mitochondrial FLVCR1b exporter. Heme export to GAPDH is associated with a temporal buildup in GAPDH-FLVCR1b association at the mitochondrial surface. Once GAPDH obtains heme from FLVCR1b it can transport it to client proteins located in cytosol or in other cellular compartments. Previous work has shown that the GAPDH client proteins are typically in complex with cell chaperone Hsp90, which is needed to drive their heme insertions in an ATP-dependent manner, with likely assistance provided by co-chaperone proteins[13,42]. Hsp90 then dissociates from the heme-replete client proteins to allow their biological function[18]. Regarding TANGO2, it was recently proposed to act either as a heme chaperone or to aid in heme transfer from heme-enriched compartments, perhaps by acting independent of other heme-transporting proteins[37,38]. We found that TANGO2 is required for mitochondrial heme transfer to GAPDH and that it associates more with FLVCR1b and less with GAPDH. On this basis, we surmise TANGO2 may function to enable FLVCR1b heme export from the mitochondria, rather than acting to chaperone heme itself or to enable GAPDH to do so. Conceivably, TANGO2 could help

FLVCR1b acquire mitochondrial heme and/or facilitate its passage of heme. A role for TANGO2 in supporting FLVCR1b function would be consistent with TANGO2 being primarily located in mitochondria[43,44], with knockout of TANGO2 expression causing the cell mitochondrial heme level to increase[38], and with TANGO2 having a poor heme binding affinity[38].

Given that other proteins may help transport heme or porphyrin through mammalian cell membranes[45,46] and that heme transfer by membrane-to-membrane contact has been invoked[41,47], it is remarkable that mitochondrial heme provision to GAPDH and its consequent deliveries to two client hemeproteins almost completely depended on FLVCR1b expression in the two cell lines that we studied. This implies that FLVCR1b plays a role in intracellular heme distribution beyond its role in enabling hemoglobin maturation during erythropoiesis as originally described by Chiabrando et al.[20]. Indeed, the existence of a FLVCR1b to GAPDH heme transfer pathway helps explain those Author's observation that FLVCR1b is needed for hemoglobin maturation, because heme delivery to apo-hemoglobin is now known to be GAPDH-dependent[10]. Likewise, our finding that FLVCR1b is required for mitochondrial heme to reach apo-sGCβ or apo-IDO1 in living cells is consistent with GAPDH being the known heme source for these two proteins[14,17]. Regarding TANGO2, our results explain an earlier report that its knockdown prevented mitochondrial heme from reaching apo-myoglobin[38], because heme provision to myoglobin also depends on GAPDH[10]. Thus, in the absence of a functional FLVCR1b exporter to provide mitochondrial heme to GAPDH, these proteins are all unable to receive heme through other cellular pathways. Indeed, FLVCR1b appears to be the main conduit that acts in conjunction with TANGO2 to supply mitochondrial heme to GAPDH for its broad distribution throughout the cell.

How heme may transfer from FLVCR1b to GAPDH can now be considered. FLVCR1b belongs to the major facilitator superfamily of transport proteins which typically consist of 12 transmembrane-spanning helices connected by hydrophilic linkers that are exposed on either side of the membrane[48,49]. These proteins fold into two domains consisting of transmembrane helices 1–6 and 7–12, which interact to form an interface that allows alternating open channels to form on either side of the membrane to facilitate small ligand passage[48]. FLVCR1b is the mitochondrially-targeted version of the larger FLVCR1a transporter and is missing the first six transmembrane helices of FLVCR1a[20]. Although a structure of FLVCR1b is not available, the structures of FLVCR1a and FLVCR2 have been solved[50–52] and by analogy, FLVCR1b has been imagined to possibly form a homodimer that would enable a similar mechanism for small ligand passage through membranes[53]. Within this context, it is fascinating that GAPDH associated with FLVCR1b on the surface of isolated mitochondria and that their heme export was stimulated by the added GAPDH. This behavior is reminiscent of how addition of the heme-binding protein hemopexin increased heme flux out of cell membrane vesicles containing FLVCR1a[54]. Given that GAPDH association with FLVCR1b increased upon stimulation of mitochondrial heme biosynthesis and then decreased as the heme loaded into the GAPDH, it is possible that their association facilitates heme export from the mitochondria. Conceivably, this could involve structural changes in FLVCR1b and GAPDH driven by their heme binding. Indeed, ligand-driven conformational effects are known to drive transport through other members of the major transporter superfamily[55], and the affinity GAPDH displays toward binding hemeproteins is inversely related to their heme contents[18]. The molecular details of the FLVCR1b-GAPDH association and its impact on mitochondrial heme export warrant further study.

Noteworthy, two proteins structurally related to FLVCR1b that were previously implicated in heme transport (FLVCR1a and FLVCR2)[56,57] have recently been found to transport choline and ethanolamine[50–52,58] to impact cell lipid composition and metabolism. TANGO2 deficiency is also known to have lipid-based impacts[43]. Given that other members of the major transporter superfamily can transport more than one type of molecule[55], a dual heme and choline/ethanolamine transport function is conceivable for FLVCR1a and FLVCR2, and whether their dual functions are related remains to be explored. Although it is currently unknown if FLVCR1b transports other molecules besides heme, FLVCR1b is conspicuously missing the amino acids that allow FLVCR1a and FLVCR2 to transport choline and ethanolamine[50,51]. Nevertheless, it will now be important to determine if FLVCR1b transports other ligands and how this may relate to mitochondrial heme export to GAPDH.

In sum, the current and previous findings suggest that heme allocation in eukaryotic cells may rely on a relatively simple pathway involving four principal proteins: Two mitochondria-associated proteins (FLVCR1b and TANGO2) that enable mitochondrial heme export to a third protein (cytosolic GAPDH) that can broadly distribute heme in cells, and a fourth protein (Hsp90) that enables GAPDH-derived heme to be inserted into numerous heme protein clients. Having this outline improves our general understanding of heme biology and can guide further investigation of the mechanisms that control heme allocation and how its dysregulation may underpin disease.

## Methods

### Ethics
The research was conducted in accordance with all applicable ethical guidelines, and the study protocol was approved by the Institutional Animal Care and Use Committee (IACUC) of the Cleveland Clinic.

### Reagents
All chemicals were purchased from Sigma unless otherwise noted. [$^{14}$C]-δ-ALA (1 μCi/μl) was purchased from ChemDepo Inc. (Camarillo, CA). FlAsH-EDT$_2$ was from Cayman Chemicals # 20704.

### Cell culture and expression of proteins by transfection
HEK293T cells (ATCC # CRL-3216), HeLa cells (ATCC # CCL-2) were provided by our in-house core cell repository at the Cleveland Clinic who had previously certified their identity and their being mycoplasma-free. The cells were cultured in Dulbecco's Modified Eagle's Medium (# 30-2002, ATCC) supplemented with 10% FBS (Gibco) at 37 °C and 5% carbon dioxide. At 60% confluence, the cells were transfected using Lipofectamine 2000 (#11668019, Invitrogen) with various mammalian expression plasmids pRK5-TC-hGAPDH-HA or the H53A variant[19], pCMV5-TC-hsGCβ(1–619)[59], pCMV3-hIDO1-FLAG (#HG11650-CF, Sino Biologicals) and hFLVCR1b_pcDNA3.1(+)-C-DYK (#U110MLFXG0, GenScript). Protein expression was allowed for 48 h.

### Transfection of siRNA and gene silencing
Expression levels of select proteins in HEK293T and HeLa cells were reduced using siRNA directed against human mRNA of FLVCR1b (Dharmacon, #CTM840033)[20]. Additionally, in HEK293T cells we used siRNA directed against human mRNAs of GAPDH (Dharmacon, #D-001830-02-20), PGRMC2 (Santa Cruz, #sc-88944), or TANGO2 (Sigma, #EHU064871) as well as scrambled versions of the siRNA's. siRNA's were used at a final concentration of 100 nM in the cell cultures along with Lipofectamine 2000. The siRNA-treated cells were cultured for 48 h before they received transfections with protein expression plasmids as described above.

### Cell supernatant preparation
HEK293T or HeLa cells were harvested using 50 mM Tris-HCl pH 7.4 buffer with 0.1% Triton X-100, 5 mM Na-molybdate and EDTA-free protease inhibitor cocktail (Roche). After three freeze-thaw cycles the lysate was centrifuged at 10,000 × $g$ for 15 min at 4 °C and the

supernatant was collected. Bradford (Bio-Rad # 500–0006) or DC method (Bio-Rad #5000111) were used to measure the protein concentration.

### Antibodies used for Western blot or immunoprecipitation

Human FLVCR1b (Novus Biologicals, #NB100-1481; dilution 1:1000), FLVCR1a (Proteintech, #26841-1-AP, dilution 1:1000), PGRMC2 (Santa Cruz, #sc-374624; dilution 1:1000), TANGO2 (Proteintech, # 27846-1-AP, dilution 1:1000), GAPDH (Cell Signaling, #2118S; dilution 1:2000), HADHA (Santa Cruz, #sc-374497; dilution 1:1000), Guanylyl Cyclase β1 (ER-19) (Sigma #G4405; dilution 1:1000), and αβ-actin (Santa Cruz, #sc-47778; dilution 1:2500), and antibodies against HA (Cell Signaling, #3724S; dilution 1:2000), FLAG (Sigma #F1804; dilution 1:1000). TOM 70 (Cell signaling, # 65619S, dilution 1:1000), and COX IV (Cell signaling, # 4844S, dilution 1:1000), Goat IgG Isotype control (Thermo Scientific, #02-6202), α-Tubulin (Proteintech, #66031-1-Ig, dilution 1:1000).

### Western blot

Proteins were detected by chemiluminescence following Western blotting using HRP conjugated secondary antibodies of anti-mouse (Bio-Rad # 170−6516; dilution 1:10,000), anti-rabbit (Biorad, #1706515; dilution 1:10,000) or anti-goat (Bio Rad #1721034; dilution 1:10,000) origin and ECL substrate (Thermo Scientific # 32106). Images were acquired and analyzed using a ChemiDoc system from Bio-Rad. The software used is BIO-RAD Image Lab Touch Software, version 3.0.1.14.

### Image analysis of Western blots

Image J quantification software (ImageJ 1.54 g; http://rsb.info.nih.gov/ij/) was used to quantify band intensities on western blots. Densitometric analysis (Image J; http://rsb.info.nih.gov/ij/) was used to measure relative protein amounts. The relative abundances of the protein of interest were determined by dividing its band intensity in each sample by the band intensity of a relevant protein control (i.e., FLVCR1b, actin) that was present and analyzed in the same sample.

### Measurement of heme

Total heme in cell supernatants was measured by an oxalic acid fluorometric method[34]. Briefly, 20 μl of cell supernatant was mixed with 980 μl of 2 M oxalic acid and boiled for 1 h. After cooling to room temperature and centrifugation, the total porphyrin was measured by its fluorescence emission at 662 nm (excitation 400 nm) relative to standard curves generated with freshly-prepared heme solutions.

Hemochromagen assay was used to measure the heme content from mitochondrial samples. Briefly, 90 μl of mitochondrial suspension was mixed with 160 μl of hemochromagen reagent (40:60 pyridine:$H_2O$, 200 mM NaOH), the heme iron was reduced by adding a few grains of sodium dithionite, the absorbance was measured in a 96-well plate at 556 nm, and was quantified by standard curve based on freshly made heme samples.

### Isolation of cell cytoplasmic and mitochondrial fractions

To prepare mitochondria-free cytosol, HEK293T cells with and without TANGO2 silencing were lysed in cytosol extraction buffer (10 mM Tris-HCl, 0.34 mM Sucrose, 3 mM $CaCl_2$, 2 mM $MgCl_2$, 0.1 mM EDTA, 1 mM DTT, 5 mM Na-molybdate and EDTA-free protease inhibitor cocktail (Roche) on ice, then centrifuged at $720 \times g$ for 5 min. The supernatant recovered was centrifuged at $10,000 \times g$ for 5 min, and then was centrifuged for 1 h in a Beckman Coulter ultracentrifuge (Optima L-100 XP) at $100,000 \times g$ and then aliquoted and stored for use. Mitochondria were isolated from HEK293T cells at 4 °C with the following procedure[20]. Cells were lysed in mitochondrial extraction buffer (0.25 M sucrose, 10 mM HEPES pH 7.4, 5 mM Na-molybdate and EDTA-free protease inhibitor cocktail (Roche). The cell suspension was passed through a 1 mL syringe and 27-gauge needle 10 times. The homogenate was centrifuged twice at $600 \times g$ for 5 min and then the supernatant was centrifuged at $10,000 \times g$ for 10 min to pellet crude mitochondria. The crude mitochondrial pellet was diluted with mitochondrial extraction buffer to a 1 mg/ml protein concentration. For IP studies the mitochondrial suspension was then solubilized with 1% Triton X-100 and protease inhibitor on ice for 30 min, centrifuged at $12,000 \times g$ for 10 min at 4 °C, and the supernatant used following protein quantification (Bio-Rad #5000111).

### Immunoprecipitation of mitochondrial samples

Solubilized mitochondrial supernatant samples (0.5 mg protein) were mixed with anti-FLVCR antibody (Novus Biologicals, #NB100-1481) and Protein G agarose beads (Cytiva # 17061801) to pull down the antibody-protein complex. Goat IgG Isotype control (Thermo Scientific, #02-6202) was used to check for specific and non-specific binding of the FLVCR antibody or beads used to capture the protein FLVCR. The beads were washed three times with 1 mL lysis buffer (50 mM Tris-HCl pH 7.4 buffer with 0.1% Triton X-100, 5 mM Na-molybdate and EDTA-free protease inhibitor cocktail) by centrifugation ($1000 \times g$ for 5 min at 4 °C). The beads were then boiled in Laemmli buffer, resolved onto 10% SDS-PAGE, and Western transferred to PVDF membrane (Bio-Rad # 1620177) to probe for proteins of interest, using the antibodies listed above.

### Live cell heme binding kinetics using FlAsH-labeled HA-TC-GAPDH

HEK293T cells in black walled 96-well plates (Greiner Bio-One, # 655090) that had been grown in heme-deficient conditions, transfected to express HA-TC-GAPDH or HA-TC-H53A GAPDH, and had or had not undergone FLVCR1b, PGRMC2, TANGO2 silencing, or rescue by transfection with FLVCR1b expression plasmid were labeled with commercially available FlAsH[19]. HeLa cells were also used to check the FLVCR1b knockdown effect on heme transfer to GAPDH, in a similar experimental condition as described above which had or had not undergone FLVCR1b silencing. Briefly, the cell monolayers were washed once with phenol red-free DMEM containing 1 g/L glucose and then given FlAsH-$EDT_2$ (Fluorescein arsenical hairpin binder-ethane dithiol) (5 μM) in Opti-MEM for 30 min at 37 °C. Afterward, the cell monolayers were washed twice with Phenol red-free DMEM containing 10% heme-depleted serum. Kinetic analyses were then performed in a SYNERGY H1 microplate reader (BioTek) using BioTek Gen5 3.12.08 software at 37 °C with excitation at 508 nm and emission at 528 nm. The fluorescence emission of FlAsH-labeled HA-TC-GAPDH in the cells was followed versus time with or without δ-ALA/Fe (1 mM delta-aminolevulinic acid and 100 μM Ferric citrate) or hemin chloride (5 μM) being added to the cells at time = 0. Kinetic data were analyzed and plotted using Origin (OriginLab Corporation, Northampton, MA, USA; version 2024b).

### Heme titration of HA-TC-GAPDH and H53A variant in cell supernatants

HEK293T cells were grown in DMEM containing 10% serum. Cells were allowed to express the HA-TC-GAPDH or H53A variant for 48 h post transfection of plasmid. Cells were then incubated with 5 μM FlAsH in Opti-MEM for 30 min at RT followed by washing with phenol red-free DMEM. Cells lysis was done using ice cold 50 mM Tris-HCl pH 7.4 buffer with 0.1% Triton X-100, 5 mM Na-molybdate, and EDTA-free protease inhibitor cocktail (Roche). The lysates were spun at 4 °C at $10,000 \times g$ for 10 min to yield supernatants for analysis. Protein concentration was measured using the Bradford method (Bio-Rad # 500–0006). 0.08 mg/ml of supernatant protein was mixed in buffer (50 mM MOPS, 150 mM NaCl, pH 7.4) at RT with varying concentrations of heme (0–2 μM), which were prepared from a stock solution of hemin chloride in DMSO. After mixing, the reaction was allowed to equilibrate for 30 min in the dark. The end point fluorescence signals

were measured in the 96-well plate reader at RT using excitation at 508 nm and emission at 528 nm.

### Heme transfer kinetics into TC-sGCβ (1– 619) using FlAsH-EDT₂

Heme transfer into FlAsH-labeled TC-sGCβ (1–619) expressed in HEK293T cells that had been grown in heme-deficient conditions and treated with corresponding siRNAs was monitored using method[59] as described above for HA-TC-GAPDH. Briefly, after FlAsH labeling and cell monolayer washing the cell fluorescence was monitored starting with or without addition of δ-aminolevulinic acid (δ-ALA) (1 mM) and ferric citrate (100 μM) at time = 0. Kinetic analyses were then performed in a SYNERGY H1 microplate reader (BioTek) at 37 °C with excitation at 508 nm and emission at 528 nm.

### Determination of $^{14}$C-labeled heme in IDO1

HEK293T cells were grown in DMEM and silenced for FLVCR1b, PGRMC2, or TANGO2 expression for 48 h before being transfected to express FLAG-IDO1. Transfected cells were then given $^{14}$C-δ-ALA (14 μM) and 10 μM ferric citrate and further cultured for 48 h. The cells were lysed using 50 mM Tris-HCl pH 7.4 buffer with 0.1% Triton X-100, 5 mM Na-molybdate, and EDTA-free protease inhibitor cocktail (Roche), and following centrifugation immunoprecipitations were done using supernatant (1 mg protein) and anti-FLAG antibody (Sigma #F1804; dilution 1:1000) and Protein G agarose beads (60 μL, Cytiva # 17061801) were used to pull down the antibody–protein complex. The beads were washed and isolated as described above and tubes containing the beads were inserted into scintillation vials which received 4 ml of scintillation fluid (Liquiscint, National Diagnostics # LS-121), and $^{14}$C counts were recorded with a scintillation counter as described[12].

### IDO1 activity assay

IDO1 activity was determined by colorimetric measurement of the accumulated L-Kynurenine product in the cell culture medium. Cells were given L-Trp (2 mM) in phenol red free DMEM containing 10% FBS with or without δ-ALA (1 mM) and ferric citrate (100 μM). After 6 h of incubation the culture medium was de-proteinized by adding an equal volume of 3% trichloroacetic acid (Sigma # T6399) and incubation at 50 °C for 30 min. After centrifugation at 9000 × $g$ for 10 min the supernatants were mixed with an equal volume of p-dimethyl-aminobenzaldehyde (Sigma # 109762; 20 mg/ml) in glacial acetic acid at room temperature and incubated 3 min to allow for chromophore formation. Absorbance at 492 nm was measured in a plate reader (Molecular Devices), and similarly processed standards containing L-Kyn (Sigma #K8625) were used to calculate the sample L-Kyn concentrations.

### Mitochondrial preparation from mouse brain and liver

Mitochondria were isolated from brain and liver obtained from 4-week-old mice (C57BL/6J,IMSR_JAX:000664, 18–22 g). The procedures were approved by IACUC of the Cleveland Clinic. Both male and female mice were used as a source, $n = 4$. Mice were sacrificed by $CO_2$ asphyxiation and cervical dislocation. The brain and liver were removed immediately and immersed in ice-cold mitochondrial isolation buffer (MIB, 0.25 M sucrose, 0.5 mM EDTA, 10 mM Tris-HCl, pH 7.4), and mitochondria were isolated by a discontinuous Percoll gradient method at 4 °C[60]. Briefly, Percoll (Sigma Aldrich #P1644) was diluted with MIB to obtain final concentrations of 12%, 26%, and 40% (v/v). Dounce homogenizers with glass pestles were used to homogenize each brain or liver sample in 12% Percoll. A gradient was prepared by pouring 26% Percoll on 40% Percoll. The homogenate in 12% Percoll was layered onto the gradient and centrifuged at 30,000 × $g$ in a Beckman Coulter ultracentrifuge (Optima L-100 XP) for 5 min. The second fraction was collected after removing the top layer and diluted 1:4 in MIB followed by centrifugation at 15,000 × $g$ for 10 min. The

pellet was resuspended in 1 ml of MIB and centrifuged at 15,000 × $g$ for 5 min. The final mitochondrial pellet was resuspended in 100 μl of MIB, its protein concentration measured, and it was then used immediately for studies.

### Localization of FLVCR1b to the outer and inner mitochondrial membranes

Proteinase K treatment: Proteinase K (Thermo Fischer Scientific, #EO0491, 5 μg) was added to the resuspended mitochondrial sample (500 μg) and the suspension was maintained by swirling every 5 min for 1 h at RT. The reaction was quenched by adding 1 μL of 100 mM PMSF into the sample. Equal total protein amounts were run on SDS-PAGE and Western blotted and developed using antibodies against Tom70, Cox4 and FLVCR1b. The FLVCR1b bands were divided by the COX4 band intensity to normalize the FLVCR1b expression.

Mitochondrial sub-fractionation: Separation of inner and outer mitochondrial membranes was performed on resuspended mitochondrial sample[25,22]. Purified mitochondria were resuspended in hypotonic medium (10 mM KCl, 2 mM HEPES, pH 7.2) with gentle stirring for 20 min on ice. Hypertonic medium (1.8 mM sucrose, 2 mM ATP, 2 mM MgSO₄, 2 mM HEPES, pH 7.2) was then added, and the solution was stirred for an additional 5 min. The mitochondrial suspensions were sonicated for 15 s at 3 amps before being layered on top of a stepwise gradient of 0.76, 1.0, 1.32, and 1.8 M sucrose and spun at 75,000 × $g$ for 3 h in an SW40 Ti rotor (Beckman). The outer membrane (OM) was collected between the 0.76 and 1.0 M interface, and the mitoplasts (MP) from the pellet. The MP and OM fractions were washed and re-pelleted (MP at 10,000 × $g$ for 10 min, OM at 120,000 × $g$ for 45 min). MP was then the inner membrane (IM). The different fractions had 4× Laemmli sample buffer added, were boiled, and then used for SDS-PAGE and Western blot analysis.

### Mitochondrial heme biosynthesis and heme labeling with $^{14}$C

Mitochondrial heme production was stimulated by resuspending mitochondria (5 mg protein) and mitochondria-free HEK293T cell cytosol (5 mg protein) to a final volume of 1 mL with incubation buffer (80 mM KCl, 50 mM MOPS, 5 mM KH₂PO₄, 1 mM EGTA, pH 7.4). This had δ-ALA (1 mM) and 4 μM ADP (Sigma #A2754) added and was then incubated for 30 min at 37 °C. In some cases, mitochondria-free cytosol from si-TANGO2 treated HEK293T cells was used, and in other cases 10 μM of $^{14}$C-δ-ALA (0.5 μCi/mg) was used. At the end of the incubations, mitochondria were pelleted by centrifugation (10,000 × $g$), washed twice, and then resuspended in incubation buffer (80 mM KCl, 50 mM MOPS, 5 mM K-phosphate, 1 mM EGTA, pH 7.4). Their heme content was quantified using the pyridine hemochromagen assay as described above[34].

### $^{14}$C-heme efflux from purified mitochondria

Heme efflux from the mitochondria isolated from mouse liver and brain was studied at room temperature[31]. The $^{14}$C-14C-heme-containing mitochondria (5 mg protein) prepared as above were incubated with or without 5 μM ADP and 20 μM of GAPDH or GST proteins for different times (0, 5, 10, 15, 30, 60 min). Reactions were stopped at each time point by transfer to ice, and the samples were then centrifuged at 4 °C at 10,000 × $g$ for 5 min. The amount of $^{14}$C-heme present in each sample supernatant and pellet was determined by scintillation counting as described previously[12]. The heme content of the supernatant was quantified using the pyridine hemochromagen assay as discussed above.

### Visualizing heme efflux from mitochondria to HA-TC-GAPDH

Wells containing FlAsH-labeled HA-TC-GAPDH protein (2 μM) and ADP (0.5 μM) in a black-walled 96-well plate at room temperature had added to them either buffer alone or containing mitochondria (0.5 mg protein) that had been reisolated after being pre-incubated with δ-ALA

and cytosols generated from either normal or TANGO2 knockdown HEK293T cells as described above. Fluorescence monitoring commenced immediately in a SYNERGY H1 microplate reader (BioTek) at room temperature with excitation at 508 nm and emission collection at 528 nm.

### Visualizing heme efflux from mitochondria to apo-TC-sGCβ

Wells in a black-walled 96-well plate containing FlAsH-labeled apo-TC-sGCβ (2 μM) and ADP (0.5 μM) at room temperature had added to them either buffer alone, buffer plus heme-loaded mitochondria (0.5 mg protein) prepared as above, or buffer containing the mitochondria and rabbit GAPDH (5 μM). Fluorescence monitoring commenced immediately in a SYNERGY H1 microplate reader (BioTek) at room temperature with excitation at 508 nm and emission collection at 528 nm.

### Proximity ligation assay (PLA) of protein-protein interactions

PLA was carried out according to the manufacturer's protocol using Duolink In Situ Detection Reagents (Sigma Aldrich, #DUO92013) to visualize the intracellular protein-protein interactions. HEK293T/HeLa cells which had or had not undergone FLVCR1b silencing were seeded onto glass coverslips immersed in 6-well plates (Genesee Scientific, # 25105) and cultured with Succinyl Acetone (SA) (400 μM) and heme-depleted serum for 2 days to deplete cell heme stores. Additionally, in FLVCR1b silenced HEK293T cells, transfection with FLVCR1b expression plasmid was done and checked for recovery in interaction between FLVCR1b and GAPDH. The coverslips were then washed and incubated with cell culture media plus δ-ALA (1 mM) and Ferric Citrate (100 μM). Coverslips were removed at different times (0, 15, 30, 45, 60 and 120 min) and were immediately fixed with 4% formaldehyde (Sigma, #F-1268), washed 3× with PBS for 5 min, and then the cells were permeabilized with 0.3% Triton X-100. Afterward, blocking was done (Sigma Aldrich, #DUO82007) followed by overnight incubations with various pairs of primary antibodies directed either against FLVCR1 (Novus Biologicals, #NB100-1481, dilution 1:100), FLVCR1 (Proteintech, #26841-1-AP, dilution 1:100), TANGO2 (Proteintech, # 27846-1-AP, dilution 1:100), GAPDH (Cell Signaling, #2118S, dilution 1:400), HADHA (Santa Cruz, #sc-374497, dilution 1:100), Hexokinase 1 (Proteintech, #19662-1-AP, dilution 1:100), Lamin B1 (Abcam, #ab16048, dilution 1:100), Lamin B1 (Fischer Scientific, #PA5-142931, dilution 1:100) or HA (Cell Signaling, #3724S; dilution 1:400). After overnight incubation the primary antibodies were washed away and PLA secondary probes were added: Anti-Rabbit MINUS (Sigma Aldrich, #DUO92005, dilution 1:5), Anti-Goat PLUS (Sigma Aldrich, #DUO92003, dilution 1:5) and Alexa-Fluor 488 anti-Mouse secondary antibody (ThermoFisher, #A-21202, dilution 1:1000), followed by 1 h incubation in a humidified chamber at 37 °C. The coverslips were twice washed in 1× Duolink In Situ wash buffer A (Sigma Aldrich, #DUO82046) for 5 min under gentle agitation. Ligation was then done by adding Ligation buffer (Sigma Aldrich, #DUO82009, dilution 1:5) and Ligase (Sigma Aldrich, #DUO82027, dilution 1:40) solutions to each sample and a further 30 min incubation in a humidified chamber at 37 °C. The coverslips were then washed twice in 1× Duolink In Situ wash buffer A for 2 min under gentle agitation, and then were placed in amplification buffer (Sigma Aldrich, #DUO82012, dilution 1:5) and polymerase (Sigma Aldrich, #DUO82028, dilution 1:80) solution and incubated 100 min at 37 °C, followed by washing twice for 10 min in 1× wash buffer B (Sigma Aldrich, #DUO82048) and then once for 1 min in 100× diluted wash buffer B, and drying at room temperature in the dark. Coverslips were individually mounted with Duolink In Situ Mounting Medium with DAPI (Sigma Aldrich, #DUO82040) onto slides, sealed with nail polish, and were imaged under a confocal microscope (Inverted Leica SP8 confocal microscope and Leica Application Suite X 3.5.7.23225 software) with an objective 63x. To determine the background PLA signal, the same procedures were followed as described above except that the cell-containing coverslips were incubated with one of the individual primary antibodies or with a non-interacting pair consisting of Laminin plus FLVCR1b, FLVCR1a, or TANGO2.

In some cases, the same PLA procedures were done for samples of mitochondria that had undergone incubation for different times with δ-ALA and cytosol, re-isolation by centrifugation twice at 10,000 × g, and placement onto poly-D-lysine coated coverslips. Microscope images for mitochondrial samples utilized a 100× objective.

### Image analysis of PLA data

In the Duolink PLA, protein-protein interactions are visualized as red dots[28]. The numbers of dots per nuclei in each image were quantified by Volocity 6.5.1 (Quorum Technologies) image analysis software. Using the compartmentalization method, two populations (Blue DAPI for nuclei and magenta dots for protein-protein interaction) were selected, and the populations in each compartment were calculated in the software. The number of dots per nuclei were averaged from each image. For each condition, experiments were done in three independent trials. Images used to determine the background PLA signal for each experimental circumstance were obtained from experiments as described above and were similarly analyzed to determine the average number of background dots per nuclei, which was reported as a relative value of 1.0 in the data presentations. The imaging results are presented as the mean of the three trial values ± standard deviation. Statistical significance (p-values) was determined using a one-way ANOVA test in the software GraphPad Prism (v9).

### Statistical analyses and reproducibility

The results are presented as the mean of the three trial values ± standard deviation. GraphPad Prism (v9) was used for the statistical analysis to measure significance (p-values) by one-way ANOVA or a two-tailed ANOVA. No statistical method was used to predetermine sample size. No data were excluded from the analyses; the experiments were not randomized; the Investigators were not blinded to allocation during experiments and outcome assessment, and due to small sample sizes animal gender was not considered in study design or analyses.

### Reporting summary

Further information on research design is available in the Nature Portfolio Reporting Summary linked to this article.

## Data availability

All data and images supporting the findings of this study are available within the main text, the Supplementary Information, and the Source Data files. Source data are provided with this paper.

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

## Acknowledgements

This work was supported by National Institutes of Health grants R01 GM130624 and R01 GM148664 (to D.J.S.) and K99 HL144921 (to E.S.), and an American Heart Association Postdoctoral Fellowship award (to E.S.). The content is solely the responsibility of the authors and does not necessarily represent the official views of the National Institutes of Health. We thank the Cleveland Clinic Lerner Research Institute Imaging Core, which provided confocal microscopy services. This work utilized the Leica SP8 confocal microscope that was purchased with funding from the National Institutes of Health SIG grant 1S10OD019972-01. We thank John Peterson for providing assistance with imaging and analysis. We thank all members of the Stuehr laboratory for helpful discussion and Dr. Nayden Naydenov (Cleveland Clinic) for providing us with mouse tissues for this study.

## Author contributions

D.J.S. conceived the study, designed the experiments, and wrote the manuscript. D.T.J. contributed to experimental design, performed the experiments, and analyzed the data. P.S. conducted selected mammalian cell experiments and performed confocal data analysis. P.B. carried out live-cell heme binding kinetics in the presence of exogenous and endogenous heme. Y.D. prepared FlAsH-labeled TC-sGCβ. E.A.S. provided input on the design of the PLA experiments and their interpretation. All authors discussed the results and contributed to the final version of the manuscript.

## Competing interests

The authors declare no competing interests.
