## [Transparent Peer Review file · Nature Communications]

Heme allocation in eukaryotic cells relies on mitochondrial heme export through FLVCR1b to cytosolic GAPDH

Corresponding Author: Dr Dennis Stuehr

Version 0:

Reviewer comments:

Reviewer #1

(Remarks to the Author)

In this paper, the authors used a cellular model and a novel GAPDH reporter construct whose heme binding in live cells can be followed by fluorescence to investigate the process of heme exit from mitochondria. They provide data suggesting that the mitochondrial protein FLVCR1b might be involved in this process. The work is potentially interesting, but the conclusions are not rigorously corroborated by the experimental findings.

MAJOR CONCERNS

1- The authors mentioned multiple times that FLVCR1b localizes in the outer mitochondrial membrane (OMM). There is no published paper demonstrating that FLVCR1b localizes in the OMM. Thus, the investigation of the mitochondrial localization of FLVCR1b is crucial to understand the potential interaction between FLVCR1b and GAPDH. To support their conclusions the authors should provide evidence that FLVCR1b resides in the OMM. In the absence of these data, they should soften their conclusions and consider additional models to explain how FLVCR1b silencing can affect heme delivery to GAPDH.

2- The authors used a siRNA to specifically downregulate FLVCR1b. Considering the strong similarities between FLVCR1a and FLVCR1b isoforms, should the authors exclude that the siRNA also affects FLVCR1a? How was the siRNA designed? It is not possible to control it, based on the information provided in the methods. I supposed the authors used a custom siRNA against FLVCR1b. If so, the authors should provide the sequence of the siRNA as well as its validation (western blot showing that the siRNA does not affect FLVCR1a). Full-unedited gels (western blot) should be provided for both FLVCR1a and FLVCR1b, and for all the other figures.

One siRNA against FLVCR1b is not sufficient to exclude off-targets. The experiments should be performed using 2 siRNA targeting the same gene.

As an additional control for the siRNA specificity, the authors should perform rescue experiments. For example, does FLVCR1b overexpression in silenced cells rescue heme transfer to TC-hGAPDH?

3- FLVCR1b siRNA restricts mitochondrial heme transfer to TC-hGAPDH. The reduced heme binding to GAPDH (Fig. 1) is convincing, however this could be an indirect effect of FLVCR1b silencing.

Experiments on isolated mitochondria (Fig. 3). In this very artificial system, the authors demonstrated that the stimulation of heme synthesis increased the PLA signal between FLVCR1b and GAPDH (see below the comments on PLA) and that GAPDH facilitated heme efflux from mitochondria but not that FLVCR1b regulated this process. This could be done by using mitochondria with reduced FLVCR1b expression.

Measurement of ¹⁴C-heme. In the experiment shown in Fig 3E-F, the authors measured ¹⁴C radioactivity in mitochondria and in the solution, not ¹⁴-C heme. To do so, they should purify labelled heme.

4- INTERACTION BETWEEN FLVCR1b and GAPDH. The authors conclude that FLVCR1b directly interacts with GAPDH based on PLA and pull-down assays. However, there are important concerns about the PLA assays:

-poor quality of the pictures. Confocal microscopy pictures, increased resolution and enlarged pictures are necessary. It is difficult/impossible to visualize the red dots. Some red dots seem over the nucleus.

-As a control for PLA, the authors used individual primary antibodies instead of a pair ("To determine the background PLA signal, the same procedures were followed as described above except that the cell-containing coverslips were incubated with only one of the individual primary antibodies instead of with a pair"). Additional controls are mandatory: It will be important to (i) perform PLA with the GAPDH-FLVCR1b antibody pairs in Flvcr1b silenced cells and (ii) performed PLA with a different pair of antibodies: GAPDH and another antibody against an unrelated protein.

-To corroborate the PLA results, the authors performed "Antibody pull down experiment". Technically speaking, this experiment is a Co-immunoprecipitation and not a pull-down assay.

The controls of the Co-IP experiment are lacking. Several controls of the Co-IP are necessary:

- Isotype control (IgG)
- to perform the same experiment in FLVCR1b silenced cells
- to perform the reverse experiment. The immunoprecipitation of GAPDH from the whole cell extract should provide a 31KDa band corresponding to FLVCR1b

It is not possible to normalize the immunoprecipitation (Fig. S3). Total protein extract, before IP, must be shown. Moreover, how did the authors separate bands of 31 and 37KDa? Did they run the immunoprecipitated proteins on separate gels? Did they blot with the same antibody used for IP?

Finally, considering that the antibody recognizes the C-term of FLVCR1, thus both FLVCR1 isoforms, it is not possible to conclude whether GAPDH interacts with FLVCR1a or FLVCR1b. Did the authors try to immunoprecipitated GAPDH and to observe whether FLVCR1a and FLVCR1b are both immunoprecipitated?

Poor description in the text / figure legends. Honestly, it is difficult to understand how the experiment was carried out by reading the text and looking at the pictures. Some information is inferred by reading the methods.

5- The authors also observed that "knockdown of FLVCR1b expression in the cells prevented GAPDH interaction with the mitochondrial surface protein HADHA..." and they concluded that "This confirmed that the changes in GAPDH interaction specifically involved its interaction with mitochondrial FLVCR1b." In my opinion, there is not a logic connection between this observation and the conclusion. The reduced PLA signal suggests that in the absence of FLVCR1b there is a reduced proximity between GAPDH and another mitochondrial protein. Which is the significance of this result? HADHA encodes the alpha subunit of the mitochondrial trifunctional protein, which catalyzes the last three steps of mitochondrial beta-oxidation of long chain fatty acids. Are there any data in the literature suggesting that GAPDH should interact with HADHA and, if so, which is the meaning of their interaction, especially in relation to FLVCR1b?

6- The H53A variants is characterized by poor heme binding affinity. Why FLVCR1b silencing should reduce the interaction between FLVCR1b and the mutant GAPDH, as observed in PLA and Co-IP experiments? The authors conclude that "This suggests that heme binding to GAPDH was coupled to the rise and fall in its interaction with FLVCR1b that occurred when cell mitochondrial heme biosynthesis is stimulated". Should the author exclude that the GAPDH mutations occurs in a site that is important for the direct binding between the two proteins? Regarding these experiments (Fig. 2E and F, Suppl Fig S4A-C), as discussed in (5), controls of PLA and Co-IP are lacking but are mandatory for the correct interpretation of the data. Thus, they should be provided.

7- The authors also describe the interaction between FLVCR1b and TANGO2. Also in this case there are important technical concerns about the experiments performed to demonstrate the interaction between FLVCR1b and TANGO2 (PLA and CoIP). Controls are lacking. Please see comments (4) regarding the interaction between FLVCR1b and GAPDH.

8- The authors discussed the potential mechanism of heme transfer from FLVCR1b to GAPDH. The authors hypothesized that FLVCR1b could dimerize to form a protein with a similar structure to FLVCR1a, working with a rocker switch mechanism to transport heme. However, recent papers reported that FLVCR1a is a choline/ethanolamine importer instead of a heme exporter, with a potential important impact on lipid metabolism. Considering that the function of FLVCR1b as a mitochondrial heme exporter was less characterized, and the sequence similarity between the two isoforms, is it possible that the observed defect in mitochondrial heme transfer to GAPDH are merely a secondary consequence of reduced mitochondrial choline uptake, or alteration in lipid metabolism and thus mitochondrial membrane composition? Of note, also TANGO2 has been involved in lipid metabolism (DOI: 10.7554/eLife.85345; doi: 10.1016/j.bbrc.2024.150047. Online ahead of print.) These issues should be discussed.

9- All the experiments have been performed in vitro, in HEK293 cells. It would be important to confirm the same data in additional cells lines, maybe cells lines that usually manage more heme, as well as in vivo.

MINOR ISSUES

10- The authors states that FLVCR1b silencing did not impairs the ability of TC-hGAPDH to accept extracellular heme. Results in Suppl. Fig 2 are not so convincing. First, in Suppl. Fig.2A the silencing of FLVCR1b is modest compared to the WB shown in Suppl Fig 1A. Also, HADAH expression is variable. Second, in Suppl. Fig.2B the authors showed two trials. It is difficult/impossible to directly compare Flvcr1b siRNA cells to control cells. At least in a trial, a small difference can be observed. Is it possible to quantify the data?

11- The description of the results and figure legends are often approximative. Few examples are reported below:

-Suppl Fig.1A: Has FLVCR1b expression been detected in the whole cells or in mitochondria samples? The author should also provide another normalization in addition to HADHA, like beta-actin, just to check whether there is no effect on mitochondrial biogenesis.

- Fig.1C legends: poor experimental details. In the figure legends "FLVCR1b knockdown inhibits the delivery of mitochondrial heme to two GAPDH client proteins in the living cells after the d-ALA/Fe addition, as judged by C it blocking the decrease in fluorescence intensity for FIAsH-labeled TC-sGCb and D it blocking the increase in the enzymatic activity and 14C heme content of IDO1". For those not expert in this kind of assay, it could be complex to note the experiments on IDO1 enzymatic activity. There is no mention of kineurine neither in the text nor in the figure legends.

- The conc of ALA and Fe citrate should be provided both in the text and in the figure legends. Which kind of medium? Heme-depleted medium?

-Suppl. Fig.1 and 7: Mitochondrial of Whole Cells protein extracts?

-Supplementary Figure 3: Upon reading the methods, I understood that the Co-IP have been performed from mitochondrial extracts. This is a relevant information that should be included at least in the figure legends. What about Suppl. Fig C?

Reviewer #2

(Remarks to the Author)

This is an excellent paper. It looks at how heme is allocated in eukaryotic cells and it establishes that movement of heme from the mitochondria to the cytosol depends on the FLVCR1b protein. TANGO2 interacts with FLVCR1b in the mitochondria, and enables its interactions with GAPDH in the cytosol. The paper substantially moves forward the understanding of the pathways for heme provision in cells. In particular, it convincingly clarifies the role of TANGO2, which to date has been a matter of confusion/disagreement in the literature. It also identifies a clear role for the FLVCR protein, which has been lurking about in the literature for many years without any real understanding of it how fits into the complex heme transport jigsaw puzzle. In this sense, Figure 5 is a major step forward in our understanding.

I ask the authors to consider the following issues.

The authors emphasise at various places (e.g. in the introduction and the discussion) that the export of mitochondrial heme by FLVCR1b to GAPDH alone is the exclusive role of FLVCR1b. GAPDH is presented as a 'fountain' supplying heme to proteins, with FLVCR1b acting as the only pipe supplying mitochondrial heme to GAPDH. While strong evidence is provided to indicate FLVCR1b as one source of mitochondrial heme to GAPDH and, in turn, to GAPDH-dependent client proteins, the authors present heme translocation to GAPDH as the sole role of FLVCR1b (and that FLVCR is the only heme source). This looked more like a leap of faith (to my reading), as it implies that all heme-binding proteins rely on GAPDH for heme loading and that FLVCR has no other roles. This is yet to be verified and cannot be generalised, despite the growing number of proteins that require GAPDH for heme delivery. So I would ask the authors to consider better justification for those wider conclusions, or adjusting their statements if necessary.

The experiments (Figs 1, 2) were performed within 2 hours after stimulation of heme biosynthesis. Can the authors exclude the existence of other back-up mechanisms, which might require longer timescales to kick-in following FLVCR1b knockdown?

Small points:

- HA-tagged is not defined, at the start of the results.

- The release of mitochondrial heme (Fig. 3) was stimulated only through the addition of GAPDH and GST. Can the authors exclude other signalling pathways?

- The confocal images in Fig. 2 were at low resolution in my copy (confocal images in supporting info look better).

- In Fig. 2F there's a steady but significant increase in the complex formation between FLVCR1b and H53A-GAPDH. The assay is only showing complex formation, not heme transfer. Is the different shape of the time-dependent trend (compared to WT-GAPDH, which peaks at 30 min and then drops) suggesting that H53 is directly involved in the formation of the complex? Is it possible that the trend is the same for both WT and H53A, but slower for H53A (so that it is only be observable much later)? A few comments by the authors on this would be appreciated.

- The word 'hemoglobinization', which I have never heard before, might need translation.

Reviewer #3

(Remarks to the Author)

In this manuscript, Thamaraparambil et al. investigate the route by which mitochondrial heme is transferred to GAPDH, an established conduit for heme allocation to recipient proteins in other cell compartments. Motivated by previous work, they identify FLVCR1b as a requisite transporter that is further supported by TANGO2. Overall, the work is straightforward, well-executed, and informative. I recommend publication following minor revisions.

1. Multiple claims are made regarding protein-protein interactions (PPIs) from their PLA assays. It is unclear to me whether this assay has the resolution to distinguish direct PPIs from indirect. For example, in Fig 2 C/D, could FLVCR1b be mediating the interaction between GAPDH and HADHA, or is it clear that the interaction is direct between the latter two? Similar for experiments in Fig. 4 with TANGO2.
2. Is the H53A GAPDH mutant expressed at comparable levels to WT?
3. The discussion states that GAPDH dissociates from mitochondria after it obtains heme. Was this shown?
4. Possible mechanisms for how TANGO2 might operate to support FLVCR1b transport should be discussed.

Reviewer #4

(Remarks to the Author)

I am typically very positive about studies coming from the Stuehr group, which has made many contributions to the heme trafficking field in eukaryotes. Although in general I am a little less enthusiastic about this manuscript, there are some interesting observations here. Basically, the study involves a mitochondrial outer membrane protein FLVCR1b, which I refer to as FLVCR for my review. FLVCR has been implicated in heme transport for many years, yet the field does not seem to be unanimous.

Here the authors use their cytoplasmic GAPDH reporters (of heme binding) to determine if FLVCR is needed for heme transport to it. They also carry out experiments to see if GAPDH and FLVCR interact. I like the approaches but I have some comments on their interpretations of results, and use of proper controls. In general, I feel that they do not use the GAPDH H53A mutant enough as control. His53 is required by GAPDH to bind heme, and short of real biochemical reconstitution experiments with purified components, it may be the best way to prove if they really are on to something.

My other general concern is the claim that they now have all the players in heme trafficking. How does heme get through the mitochondrial inner membrane, or the intermembrane space to FLVCR? Why do their results say "this completes an outline of the entire process" as they state in the Discussion?

Specific comments:

1. "Our results identify a transporter protein located in the outer mitochondrial membrane, Feline Leukemia Virus subgroup C Receptor 1b (FLVCR1b), as the sole conduit for mitochondrial heme export to GAPDH in the cells, operating through a mechanism that involves a direct FLVCR1b-GAPDH interaction for heme exchange. "

Not sure if this claim is proven, particularly in view of the outer membrane association of FLVCR1b. As I state above, how does heme get from the ferrochelatase in side mitos to FLVCR1b?

2. One aspect that is hard for me to understand, based on the authors statement: "The targeted siRNA treatment diminished cell FLVCR1b protein expression relative to a scrambled siRNA control by 70 +/- 9% (n = 5) (Fig. 1A and S1A) and this did not impact the cell heme level from increasing in response to the d-ALA/Fe addition (Table S1), consistent with a previous report. "

So the FLVCR knockdown reduces expression by about 70% but the knockdown line/cell has the same levels of heme. Where is this heme, and how did it get there? Also, when ALA is added to induce heme levels ~ 3 fold, the knockdown prevents 90% of heme going to GAPDH.

3. Fig 1D, why is there such low heme (C14 labelled) in the "no ALA" cells and it appears tens of fold more in the +ALA siFLVCR line?

4. "Antibody pulldown experiments were done to independently assess the PLA results and they confirmed there is a temporal buildup in GAPDH-FLVCR1b association that peaked at 30 min after the d-ALA/Fe addition (Fig S3A and B). " I know that the quantitations show some increase at the 30 min timepoint but looking at the western itself, it seems underwhelming.

5. "In separate PLA-based experiments, we found that knockdown of FLVCR1b expression in the cells prevented GAPDH interaction with the mitochondrial surface protein HADHA and the increase in their interaction that otherwise occurred in response to d-ALA/Fe addition in the control cells (Fig. 2C and D). This confirmed that the changes in GAPDH interaction specifically involved its interaction with mitochondrial FLVCR1b."

Can this section be clarified? What is HADHA, why does it interact with GAPDH, why does siFLVCR reduce it? How does this show a direct interaction of GAPDH with FLVCR?

6. It appears that Fig2E somewhat contradicts the direct interaction (handoff of heme) from FLVCR to GAPDH: A His53 variant of GAPDH that does not bind heme still shows an increase in interaction with FLVCR upon ALA addition, albeit after 30 min (120min) rather than at 30 min. (The pulldowns in Supp Fig 4 also suggest the interaction is independent of heme). Their conclusion is thus hard to rationalize: "This suggests that heme binding to GAPDH was coupled to the rise and fall in its interaction with FLVCR1b that occurred when cell mitochondrial heme biosynthesis is stimulated. "

7. Results in Fig 3 show transfer of heme to GAPDH and direct interaction with FLVCR. These experiments would be strengthened by using the GAPDH His53Ala variant as a control.

8. Seems like results in Fig1D and 4C are nearly identical, yet different experiments. Is this just coincidence or maybe a mix-up?

9. In Discussion the authors state: "Our current findings clarify its role by showing TANGO2 interacts with FLVCR1b and not with GAPDH and enables FLVCR1b export of mitochondrial heme to GAPDH. This role is consistent with knockout of

TANGO2 causing an increase in the cell mitochondrial heme level 31, with TANGO2 displaying a poor affinity toward binding heme 31, and with a recent study that cast doubt on its functioning in intracellular heme transport 36. " I am not sure how their data is consistent with both ref31 and ref 36, or how it clarifies the situation of TANGO2 as a heme chaperone. It would also help to know when they say it is in the mito, where is it (e.g. matrix, IMS, IM, OM)?

Reviewer #5

(Remarks to the Author)

"I co-reviewed this manuscript with one of the reviewers who provided the listed reports. This is part of the Nature Communications initiative to facilitate training in peer review and to provide appropriate recognition for Early Career Researchers who co-review manuscripts."

Version 1:

Reviewer comments:

Reviewer #1

(Remarks to the Author)

The authors made significant efforts to enhance the quality of the manuscript. We appreciate their work in improving the quality of the PLA images, refining the PLA controls, and validating the FLVCR1b-GAPDH and the FLVCR1b-TANGO2 interactions. The Co-IP is fundamental to demonstrate the existence of an interaction between proteins because PLA permits detection of proximity between proteins (at distances < 40 nm). Thus, considering the relevance of this experiment, the Co-IP results (Fig. S9 and Fig. S14) should be placed among the main figures. Furthermore, the graphical representation of the Co-IP results should be improved to ensure more clarity. It should be clearly understood which protein has been immunoprecipitated and which one has been immunoblotted by simply observing the picture. For example, the graphical representation will be improved by adding a note on the top of Fig.S9A to show that the first 4 lanes on the left are the IgG isotype control and the 4 lanes on the right are the FLVCR1b IP. Same suggestion for Fig.S14. Please refer to published IP representation to improve the clarity of your picture. Concerning Fig.S9, why are IgG bands visible only in the IgG isotype controls and not where FLVCR1b has been immunoprecipitated? We cannot find information about the type of IgG in the methods. Which IgG has been used for control?

Fig.S9E: this is another essential control of the specificity of the GAPDH-FLVCR1b interaction. Unfortunately, the resolution of the image is poor and GAPDH is difficult to detect. We recommend improving the picture. Also in this case, at a glance the image seems like a simple western blot. It would be better to represent the picture as a co-IP.

Fig. S14: is it possible to improve the resolution of Fig. S14D?

Supplementary Figure S3. "Effect of FLVCR1b knockdown on the ability of FIAsH-TChGAPDH expressed in cells to bind exogenously-supplied versus endogenous heme". This is a very interesting experiment that should be emphasized in a main figure.

Minor comments:

- In the legend of Fig.S3 we read "vehicle plus heme (5 μ M), or dALA + Fe (1mM and 100 μ M) as indicated". The picture reporter FeCit whereas in the legend we can find only Fe. What does it mean "as indicated"? Maybe a clearer way to write it could be 1mM dALA + 100 μ M FeCit?
- The authors used two different antibodies to detect FLVCR1a and FLVCR1b. Please indicate in the Methods section which antibody was used for each protein.
- Fig. S1C: is it possible to improve the resolution of the picture? Maybe it is sufficient to modulate the intensity/saturation of the signal.
- I cannot find information about the plasmid used to express Flvcr1b in the methods.
- Please uniform text and figures: sometimes d-ALA/Fe sometimes d-ALA/FeCit.

English revision is needed. Below please find some suggestions:

- Lanes 70-71: "We also show that the protein TANGO2, likely through it interacting with FLVCR1b, is needed for the mitochondrial heme transfer to GAPDH". Change the underlying text with "through its interaction".
- Lane 85: "We then utilized an siRNA". Change an with a.
- Lane 85: "We then utilized an siRNA that can specifically knock down expression of FLVCR1b but otherwise was reported not to cause off-target effects 20 to determine how FLVCR1b knockdown would impact heme loading into TC-hGAPDH in response to our stimulating cell mitochondrial heme biosynthesis by providing the heme precursor molecules d-aminolevulinic acid (d-ALA) and ferric citrate, which in our hands typically causes the heme level in HEK293T cells and the level of heme bound in the TChGAPDH to increase by about 3-fold within 2 h". This sentence is too long. Please rephrase the sentence to improve clarity and readability.
- Lane 92-98: This sentence is too long. Please rephrase the sentence to improve clarity and readability.
- Lane 141: "A dual membrane location for FLVCR1b is consistent with it functioning to export heme from mitochondria to cytosolic GAPDH". Change it with its
- Lane 145: We next investigated if mitochondrial heme export to GAPDH involved it interacting with FLVCR1b by utilizing

the Duolink proximity ligation assay (PLA). Correct the grammar. I suppose Duolink is a registered trademark so it should be written as Duolink ®.

Reviewer #2

(Remarks to the Author)

The authors have made very extensive changes to the manuscript, including a lot of new data and the removal of the His53 data which was raised by more than one reviewer. I am happy with these changes.

Reviewer #3

(Remarks to the Author)

I am satisfied with the responses the authors have offered for my original questions. I have no further concerns to raise.

Reviewer #5

(Remarks to the Author)

Version 2:

Reviewer comments:

Reviewer #1

(Remarks to the Author)

The authors have addressed all our concerns.

Reviewer #5

(Remarks to the Author)

Dear Editors:

Thank you for obtaining 5 helpful reviews of our manuscript “**Heme allocation in eukaryotic cells relies on mitochondrial heme export through FLVCR1b to cytosolic GAPDH**” by Drs. Jayaram *et al* and Stuehr. As requested, we have performed extensive additional studies, increased manuscript clarity, and modified certain claims, as is described in our replies to the Reviewer comments. Together, we believe this has significantly improved and strengthened the study and we thank the Editors and Reviewers for enabling this process.

Reviewer #1 (Remarks to the Author):

In this paper, the authors used a cellular model and a novel GAPDH reporter construct whose heme binding in live cells can be followed by fluorescence to investigate the process of heme exit from mitochondria. They provide data suggesting that the mitochondrial protein FLVCR1b might be involved in this process. The work is potentially interesting, but the conclusions are not rigorously corroborated by the experimental findings.

We are glad the Reviewer found our study interesting. We have now improved the rigor of the study as explained below.

MAJOR CONCERNS

1- The authors mentioned multiple times that FLVCR1b localizes in the outer mitochondrial membrane (OMM). There is no published paper demonstrating that FLVCR1b localizes in the OMM. Thus, the investigation of the submitochondrial localization of FLVCR1b is crucial to understand the potential interaction between FLVCR1b and GAPDH. To support their conclusions the authors should provide evidence that FLVCR1b resides in the OMM. In the absence of these data, they should soften their conclusions and consider additional models to explain how FLVCR1b silencing can affect heme delivery to GAPDH.

We thank the reviewer for alerting us about this. The report on FLVCR1b by Tolosano *et al* showed that it is localized to the mitochondria, but it did not report its submitochondrial localization. We now include new data we obtained using two established approaches that show FLVCR1b is present in both the outer and inner mitochondrial membranes. In one approach, isolated mitochondria were treated with proteinase K to determine if our protein of interest is in the outer membrane (where it would be protease susceptible). Control proteins that are known to be localized either

on the outer membrane (TOM 70) or the inner membrane (COX4) served as controls. We found that the proteinase K treatment diminished the FLVCR1b protein content of the mitochondria by 70%, consistent with its helix-connecting loops being solvent exposed and it being present in the outer membrane. As a second approach, we performed mitochondrial sub-fractionation to see where FLVCR1b is localized compared to standard mitochondrial marker proteins noted above. This showed that FLVCR1b is present in both the outer and inner mitochondrial membrane fractions. These new data are shown in Fig. S5.

2- The authors used a siRNA to specifically downregulate FLVCR1b. Considering the strong similarities between FLVCR1a and FLVCR1b isoforms, should the authors exclude that the siRNA also affects FLVCR1a? How was the siRNA designed? It is not possible to control it, based on the information provided in the methods. I supposed the authors used a custom siRNA against FLVCR1b. If so, the authors should provide the sequence of the siRNA as well as its validation (western blot showing that the 1b siRNA knocks down 1b but does not affect FLVCR1a).

We have clarified by adding the requested siRNA information into the revised manuscript. We also state that we utilized the exact same siRNA to diminish FLVCR1b expression that was characterized and utilized in the Tolosano paper. These authors designed an siRNA against the 5' UTR of FLVCR1b which should not affect expression of FLVCR1a. As quoted from their report, "quantitative RT-PCR (qRT-PCR) analysis confirmed a reduction of Flvcr1b mRNA 72 hours after siRNA transfection, with no effect on Flvcr1a mRNA (Supplemental Figure 6)". Nevertheless, we have now tested its specificity in our system and confirmed by Western that the siRNA only diminished FLVCR1b and not FLVCR1a protein expression in our system. We have added this new information into the revised as Fig. S1 C&D. Thus, the specificity of the siRNA that we used is now demonstrated.

Full-unedited gels (western blot) should be provided for both FLVCR1a and FLVCR1b, and for all the other figures.

The full WB are now included with the revised as a separate PowerPoint file.

One siRNA against FLVCR1b is not sufficient to exclude off-targets. The experiments should be performed using 2 siRNA targeting the same gene.

The original Tolosano study did check for off-target effects of the FLVCR1b siRNA and they found it did not alter mitochondrial expression of cytochrome c oxidase or diminish cell heme content. We observed similar specificity in that the siRNA treatment had no effect on the cell's ability to express several proteins of interest (FLVCR1a, IDO, sGC β , HA-GAPDH) and did not alter cell heme content. We believe that their published results along with our observations obviate the need for us to repeat the study using a second

siRNA for FLVCR1b.

As an additional control for the siRNA specificity, the authors should perform rescue experiments. For example, does FLVCR1b overexpression in silenced cells rescue heme transfer to TC-hGAPDH?

We have performed this important control experiment and now include the results in the revised Fig. 1A and S1A. Specifically rescuing FLVCR1b expression in the FLVCR1b knockdown cells did fully rescue mitochondrial heme transfer to TC-GAPDH (New Fig. 1B). This suggests that the only significant effect of the siRNA treatment was to change the FLVCR1b expression level in the cells, which in turn impacted cell heme allocation.

3A- FLVCR1b siRNA restricts mitochondrial heme transfer to TC-hGAPDH. The reduced heme binding to GAPDH (Fig. 1) is convincing, however this could be an indirect effect of FLVCR1b silencing.

We included in the original MS measurements of the cell total heme level that indicate the siRNA-mediated knockdown of FLVCR1b expression does not alter the cell total heme levels. This eliminates a possible mechanism where the knockdown causes altered cell heme. Also, our new results show that FLVCR1b re-expression (rescue) in the knock-down cells can reverse the negative effect of the siRNA on mitochondrial heme transfer to TC-GAPDH. In addition, we show in Fig. S3 that when we added external heme to the FLVCR1b-knockdown cells, their TC-GAPDH could normally bind the heme. Together, this provides reasonable assurance that alternative mechanisms preventing TC-GAPDH from binding mitochondrial heme in the FLVCR1b knockdown cells are not in play.

3B- Experiments on isolated mitochondria (Fig. 3). In this very artificial system, the authors demonstrated that the stimulation of heme synthesis increased the PLA signal between FLVCR1b and GADPH (see below the comments on PLA) and that GADPH facilitated heme efflux from mitochondria but not that FLVCR1b regulated this process. This could be done by using mitochondria with reduced FLVCR1b expression.

We agree that this could provide additional insight. However, Tolosano reported one cannot generate a cell or mouse that only has its FLVCR1b knocked out (i.e., they would also have FLVCR1a knocked out). This leaves siRNA knockdown as the only approach. But due to financial considerations we are unable to purchase enough siRNA to silence the number of cultured cells that would be needed to provide sufficient isolated mitochondria to perform the suggested experiment.

3C- Measurement of 14C-heme. In the experiment shown in Fig 3E-F, the authors measured 14C radioactivity in mitochondria and in the solution, not 14-C heme. To do so, they should purify labelled heme.

As requested, to check if the measured 14C counts are heme-based, we have now performed spectroscopic heme measurements on the samples using the hemochromogen method, which generates a chromophore that is specific for heme. Results indicate that the increases in sample 14C counts correlate with increased sample heme as measured by this spectroscopic method. This is now reported in Table S2.

4A- INTERACTION BETWEEN FLVCR1b and GAPDH. The authors conclude that FLVCR1b directly interacts with GAPDH based on PLA and pull-down assays. However, there are important concerns about the PLA assays:

-poor quality of the pictures. Confocal microscopy pictures, increased resolution and enlarged pictures are necessary. It is difficult/impossible to visualize the red dots. Some red dots seem over the nucleus.

We have now included images with greater pixilation and at greater magnifications in which the red dots are clearly visible. We now present these images in Figs 2,4D and S6, S7, S8,S10, S14, S15 S16 and S17 along with the control images, as was done in a recent report by Tolosano's group that used PLA to study the interactions of FLVCR1a with IP3R2-VDAC (*Cell Rep.* 2024 43(3):113854. doi: 10.1016/j.celrep.2024.113854).

4B-As a control for PLA, the authors used individual primary antibodies instead of a pair ("To determine the background PLA signal, the same procedures were followed as described above except that the cell-containing coverslips were incubated with only one of the individual primary antibodies instead of with a pair"). Additional controls are mandatory: It will be important to (i) perform PLA with the GAPDH-FLVCR1b antibody pairs in Flvcr1b silenced cells and (ii) performed PLA with a different pair of antibodies: GAPDH and another antibody against un unrelated protein.

We have now performed the two requested experiments and have added the data into the revised manuscript. (i) New PLA data using the GAPDH-FLVCR1b antibody pairs in FLVCR1b silenced cells is now Fig. 2C & D. It shows that there is a near-background PLA signal for the FLVCR1b-GAPDH in the FLVCR1b silenced cells, and that rescuing these cells by FLVCR1b transfection restores the pattern of PLA interaction we see in normal cells or in cells receiving scrambled siRNA. (ii) New control PLA data using different pairs of antibodies as suggested are now in Fig. S6. It involves FLVCR1b and laminin, as used recently by Tolosano et al in the *Cell Rep.* article noted above.

4C-To corroborate the PLA results, the authors performed “Antibody pull down experiment”. Technically speaking, this experiment is a Co-immunoprecipitation and not a pull-down assay.

We have re-worded to use co-IP.

The controls of the Co-IP experiment are lacking. Several controls of the Co-IP are necessary:

- Isotype control (IgG)

We now include this control. It shows that the FLVCR1b co-IP is due to the specific Ab and not the IgG. For the IgG controls, we have used Goat IgG Isotype control and checked for specific and non-specific binding of the antibody or beads used to capture the protein FLVCR1b. Our IgG control results ensured that any detected protein interaction is specific to FLVCR1b and not due to random protein adhesion to the beads.

- to perform the same experiment in FLVCR1b silenced cells

We have now performed this co-IP and show the Westerns in Fig. S9. It shows much less FLVCR1b protein from the silenced cells.

- to perform the reverse experiment. The immunoprecipitation of GAPDH from the whole cell extract should provide a 31KDa band corresponding to FLVCR1b.

We agree that this complementary approach could be useful, but in our case it is not feasible because the cells have too much GAPDH and too little FLVCR1b expression, and so at any given time, very little of the total cell GAPDH is interacting with FLVCR1b.

It is not possible to normalize the immunoprecipitation (Fig. S3). Total protein extract, before IP, must be shown.

We now include the input sample Western results in the same Figure S9.

Moreover, how did the authors separate bands of 31 and 37KDa? Did they run the immunoprecipitated proteins on separate gels? Did they blot with the same antibody used for IP?

Yes, we used the same Ab as for the co-IP. We first imaged FLVCR1b, then stripped and re-probed for GAPDH or for HA using Abs against those proteins.

Finally, considering that the antibody recognizes the C-term of FLVCR1, thus both

FLVCR1 isoforms, it is not possible to conclude whether GAPDH interacts with FLVCR1a or FLVCR1b. Did the authors try to immunoprecipitated GAPDH and to observe whether FLVCR1a and FLVCR1b are both immunoprecipitated?

We did not co-IP using our GAPDH Ab for the reasons noted above, and that it may interact with both FLVCR1a and 1b. Also, while it is true our FLVCR1 Ab recognizes both 1a and 1b, for our immunoprecipitations this is not a concern, because we only worked with isolated mitochondrial preparations, so very little of no FLVCR1a was present in such samples (It is primarily located in the cell membrane).

For our whole cell PLA experiments, please note that we have added new data into the revised manuscript (Fig. 2 C& D) showing that in the cells whose FLVCR1b expression is specifically silenced but otherwise still express a normal level of FLVCR1a, we observed that after dALA/Fe addition there was no increase in the PLA signal for FIVCR1-GAPDH interaction by 30 min. This indicates that the PLA signal gain that we otherwise do see in the non-knockdown cells at 30 min does not involve a GAPDH interaction with cell FLVCR1a, but instead involves a GAPDH interaction with FLVCR1b. We also added new data (Fig. S8) from PLA experiments that used an Ab directed specifically against FLVCR1a, and again we see no signal gain for the FLVCR1a-GAPDH interaction at 0 & 30 min after the dALA/Fe addition. Interestingly, we did see some increase in their interaction at 120 min after d-ALA/Fe addition. This suggests that GAPDH may eventually interact with FLVCR1a in such cells, but after its more immediate interaction with FLVCR1b has already taken place. In any case, we can now conclude that the increase and subsequent decrease in the PLA signal that occurs within a 1 h period after dALA/Fe addition to cells is specifically due to changes in the GAPDH interaction with FLVCR1b and not with FLVCR1a.

Poor description in the text / figure legends. Honestly, it is difficult to understand how the experiment was carried out by reading the text and looking at the pictures. Some information is inferred by reading the methods.

We regret this confusion and have increased the information content of the Figure legends.

5- The authors also observed that “knockdown of FLVCR1b expression in the cells prevented GAPDH interaction with the mitochondrial surface protein HADHA....” and they concluded that “This confirmed that the changes in GAPDH interaction specifically involved its interaction with mitochondrial FLVCR1b.” In my opinion, there is not a logic connection between this observation and the conclusion. The reduced PLA signal suggests that in the absence of FLVCR1b there is a reduced proximity between GAPDH

and another mitochondrial protein. Which is the significance of this result? HADHA encodes the alpha subunit of the mitochondrial trifunctional protein, which catalyzes the last three steps of mitochondrial beta-oxidation of long chain fatty acids. Are there any data in the literature suggesting that GAPDH should interact with HADHA and, if so, which is the meaning of their interaction, especially in relation to FLVCR1b?

In retrospect HADHA was not an optimal choice because it is reported to be located in the mitochondrial inner membrane. We therefore have repeated the PLA experiments that look at the GAPDH-mitochondrial interaction using an antibody against a known mitochondrial outer membrane surface protein, Hexokinase 1. The new results (Fig. S10) show that FLVCR1b knockdown reduces the PLA signal for GAPDH interaction with Hexokinase 1 both before and after adding dALA/Fe to the cells. This suggests that FLVCR1b is needed for GAPDH to interact with the mitochondrial surface.

6- The H53A variants is characterized by poor heme binding affinity. Why FLVCR1b silencing should reduce the interaction between FLVCR1b and the mutant GAPDH, as observed in PLA and Co-IP experiments? The authors conclude that “This suggests that heme binding to GAPDH was coupled to the rise and fall in its interaction with FLVCR1b that occurred when cell mitochondrial heme biosynthesis is stimulated”. Should the author exclude that the GAPDH mutations occurs in a site that is important for the direct binding between the two proteins?

We agree, at present we do not know if H53A GAPDH may also have a defective mitochondrial interaction. We therefore decided to remove the H53A data from the revised manuscript. This does not alter the main findings or conclusions in our study.

Regarding these experiments (Fig. 2E and F, Suppl Fig S4A-C), as discussed in (5), controls of PLA and Co-IP are lacking but are mandatory for the correct interpretation of the data. Thus, they should be provided. (Rev means discussed in 4)

These comments refer to the H53A GAPDH data which are no longer in the revised manuscript.

7- The authors also describe the interaction between FLVCR1b and TANGO2. Also in this case there are important technical concerns about the experiments performed to demonstrate the interaction between FLVCR1b and TANGO2 (PLA and CoIP). Controls are lacking. Please see comments (4) regarding the interaction between FLVCR1b and GAPDH.

Appropriate controls for the PLA and co-IP experiments have now been included in Fig. S14.

8- The authors discussed the potential mechanism of heme transfer from FLVCR1b to GAPDH. The authors hypothesized that FLVCR1b could dimerize to form a protein with a similar structure to FLVCR1a, working with a rocker switch mechanism to transport heme. However, recent papers reported that FLVCR1a is a choline/ethanolamine importer instead of a heme exporter, with a potential important impact on lipid metabolism. Considering that the function of FLVCR1b as a mitochondrial heme exporter was less characterized, and the sequence similarity between the two isoforms, is it possible that the observed defect in mitochondrial heme transfer to GAPDH are merely a secondary consequence of reduced mitochondrial choline uptake, or alteration in lipid metabolism and thus mitochondrial membrane composition? Of note, also TANGO2 has been involved in lipid metabolism (DOI: 10.7554/eLife.85345; doi: 10.1016/j.bbrc.2024.150047. Online ahead of print.) These issues should be discussed.

We agree that this could be important contextual information for readers, so we now mention these distinct transport roles for FLVCR1a and TANGO2 proteins in the discussion. There currently is no published evidence that FLVCR1b acts to transport choline or related biomolecules. Instead, the Tolosano group has published strong evidence that FLVCR1b is localized to the mitochondria and involved in its heme trafficking. Regarding FLVCR1a, Doty et al have previously published evidence it acts as a cell membrane heme exporter. Regarding TANGO2, a recent publication in Nature describes its function in mitochondrial heme trafficking. Our current data further support heme trafficking roles for FLVCR1b and TANGO2. While it is conceivable that FLVCR1b and TANGO2 could transport other small molecules like choline, such transport functions need not impact their heme trafficking or be mutually exclusive.

9- All the experiments have been performed in vitro, in HEK293 cells. It would be important to confirm the same data in additional cell lines, maybe cells lines that usually manage more heme, as well as in vivo.

We have now repeated the core experiments in a different human cell line (HeLa) and include the new results in Fig. S2, S6C and D, and , S7, S10 C and D. The findings using HeLa are essentially identical to what we observed when using the HEK293 cells, and thus suggest a fundamental mechanism in cell biology. This would be consistent with GAPDH-dependent mitochondrial heme trafficking operating in different mammalian cell lines and even in yeast.

Regarding animal studies, we agree they could conceivably add value, but as noted above the Tolosano group reported that one cannot make a FLVCR1b-specific KO mouse and that the pan FLVCR1 (a & b) knockout is lethal. The same holds for a GAPDH knockout. Our future work aims to identify the molecular details of the GAPDH-FLVCR1b interaction, which could then provide a more precise means to block their

heme exchange in live animals.

MINOR ISSUES

10- The authors states that FLVCR1b silencing did not impairs the ability of TC-hGAPDH to accept extracellular heme. Results in Suppl. Fig 2 are not so convincing. First, in Suppl. Fig.2A the silencing of FLVCR1b is modest compared to the WB shown in Suppl Fig 1A. Also, HADAH expression is variable. Second, in Suppl. Fig.2B the authors showed two trials. It is difficult/impossible to directly compare Flvcr1b siRNA cells to control cells. At least in a trial, a small difference can be observed. Is it possible to quantify the data?

We have repeated the experiment and show the results in Figure S3. The data clearly show that FLVCR1b knockdown occurred in the cells and that this greatly diminished mitochondrial heme transfer to TC-GAPDH but did not inhibit TC-GAPDH from obtaining heme that was provided exogenously to the cell cultures.

11- The description of the results and figure legends are often approximative. Few examples are reported below:

-Suppl Fig.1A: Has FLVCR1b expression been detected in the whole cells or in mitochondria samples?

Due to the relatively low FLVCR1b expression level and numbers of mitochondria in cells, we could reliably detect FLVCR1b expression by Western only in crude or purified mitochondrial samples. This is now noted in the revised.

The author should also provide another normalization in addition to HADHA, like beta-actin, just to check whether there is no effect on mitochondrial biogenesis.

When working with crude or purified mitochondria samples, we could only use mitochondrial proteins like HADHA as expression controls. Regarding an effect of FLVCR1b knockdown on mitochondrial biogenesis, Tolosano et al had reported that FLVCR1b knockdown did not change the level of mitochondrial cytochrome c oxidase present in their cell preparations, which implies it did not greatly alter cell mitochondrial numbers.

- Fig.1C legends: poor experimental details. In the figure legends “FLVCR1b knockdown inhibits the delivery of mitochondrial heme to two GAPDH client proteins in the living cells after the d-ALA/Fe addition, as judged by C it blocking the decrease in fluorescence intensity for FIAsH-labeled TC-sGCb and D it blocking the increase in the

enzymatic activity and 14C heme content of IDO1". For those not expert in this kind of assay, it could be complex to note the experiments on IDO1 enzymatic activity. There is no mention of kineurine neither in the text nor in the figure legends.

Thank you for noticing this. We now provide an explanation.

- The conc of ALA and Fe citrate should be provided both in the text and in the figure legends. Which kind of medium? Heme-depleted medium?

We now provide an explanation. δ -ALA (1 mM) and ferric citrate (100 μ M). Heme depleted medium was used.

-Suppl. Fig. 1 and 7: Mitochondrial of Whole Cells protein extracts?

We now provide an explanation. Fig S1 is mitochondrial samples and Fig. S7 is whole cell protein extracts.

-Supplementary Figure 3: Upon reading the methods, I understood that the Co-IP have been performed from mitochondrial extracts. This is a relevant information that should be included at least in the figure legends. What about Suppl. Fig C?

We now provide an explanation. Mitochondrial extracts were used throughout.

Reviewer #2 (Remarks to the Author):

This is an excellent paper. It looks at how heme is allocated in eukaryotic cells and it establishes that movement of heme from the mitochondria to the cytosol depends on the FLVCR1b protein. TANGO2 interacts with FLVCR1b in the mitochondria, and enables its interactions with GAPDH in the cytosol. The paper substantially moves forward the understanding of the pathways for heme provision in cells. In particular, it convincingly clarifies the role of TANGO2, which to date has been a matter of confusion/disagreement in the literature. It also identifies a clear role for the FLVCR protein, which has been lurking about in the literature for many years without any real understanding of it how fits into the complex heme transport jigsaw puzzle. In this sense, Figure 5 is a major step forward in our understanding.

Thank you for highlighting these advances.

I ask the authors to consider the following issues.

The authors emphasise at various places (e.g. in the introduction and the discussion) that the export of mitochondrial heme by FLVCR1b to GAPDH alone is the exclusive role of FLVCR1b. GAPDH is presented as a 'fountain' supplying heme to proteins, with FLVCR1b acting as the only pipe supplying mitochondrial heme to GAPDH. While strong evidence is provided to indicate FLVCR1b as one source of mitochondrial heme to GAPDH and, in turn, to GAPDH-dependent client proteins, the authors present heme translocation to GAPDH as the sole role of FLVCR1b (and that FLVCR is the only heme source). This looked more like a leap of faith (to my reading), as it implies that all heme-binding proteins rely on GAPDH for heme loading and that FLVCR has no other roles. This is yet to be verified and cannot be generalised, despite the growing number of proteins that require GAPDH for heme delivery. So I would ask the authors to consider better justification for those wider conclusions, or adjusting their statements if necessary.

We have adjusted our statements as suggested. FLVCR1b may certainly have additional roles and alternative heme transfer pathways may be present in cells.

The experiments (Figs 1, 2) were performed within 2 hours after stimulation of heme biosynthesis. Can the authors exclude the existence of other back-up mechanisms, which might require longer timescales to kick-in following FLVCR1b knockdown?

We can't exclude that other FLVCR1b-independent heme transport pathways may become active in the knockdown cells at times beyond what we followed (2 h and 6 h). We now note this possibility in the revised.

Small points:

- HA-tagged is not defined, at the start of the results.

Done.

- The release of mitochondrial heme (Fig. 3) was stimulated only through the addition of GAPDH and GST. Can the authors exclude other signalling pathways?

Our current data does not exclude that other macromolecules present in cytosol might trigger a release of mitochondrial heme. Investigating this is beyond the scope of our current study. We chose GST to compare to GAPDH because it had been one of a handful of cell proteins proposed over the years to possibly act as an intracellular heme chaperone. It would be interesting to compare more macromolecules, and we plan to do so.

- *The confocal images in Fig. 2 were at low resolution in my copy (confocal images in supporting info look better).*

We now include better images.

- *In Fig. 2F there's a steady but significant increase in the complex formation between FLVCR1b and H53A-GAPDH. The assay is only showing complex formation, not heme transfer. Is the different shape of the time-dependent trend (compared to WT-GAPDH, which peaks at 30 min and then drops) suggesting that H53 is directly involved in the formation of the complex? Is it possible that the trend is the same for both WT and H53A, but slower for H53A (so that it is only be observable much later)? A few comments by the authors on this would be appreciated.*

Yes, either or both scenarios are possible. Based on the H53A comments by the Reviewers, we've realized that the data we have for H53A GAPDH is not easily interpretable and more work will be needed before we can understand the differences in its interaction relative to wild type GAPDH. Thus, we have omitted the H53A data in the revised.

- *The word 'hemoglobinization', which I have never heard before, might need translation.*

We have simplified it to hemoglobin formation for better general understanding. The WWW defines "hemoglobinization" as The formation or concentration of hemoglobin, and the term has been used in the literature: Clin Lab. 2006;52(3-4):107-14. "Hemoglobinization and functional availability of iron for erythropoiesis in case of thalassemia and iron deficiency anemia"

Reviewer #3 (Remarks to the Author):

In this manuscript, Thamaraparambil et al. investigate the route by which mitochondrial heme is transferred to GAPDH, an established conduit for heme allocation to recipient proteins in other cell compartments. Motivated by previous work, they identify FLVCR1b as a requisite transporter that is further supported by TANGO2. Overall, the work is straightforward, well-executed, and informative. I recommend publication following minor revisions.

1. Multiple claims are made regarding protein-protein interactions (PPIs) from their PLA assays. It is unclear to me whether this assay has the resolution to distinguish direct

PPIs from indirect. For example, in Fig 2 C/D, could FLVCR1b be mediating the interaction between GAPDH and HADHA, or is it clear that the interaction is direct between the latter two? Similar for experiments in Fig. 4 with TANGO2.

It is broadly accepted that the PLA can detect two proteins (antigens) that are within 40 nm or less of one another. The maximal distance is not short enough to definitively demonstrate that two protein antigens are engaged in a direct physical interaction, but it is sufficient to support a very close 'proximity', for example, within a complex. Thus, it could involve proteins that are close in a multi-protein complex but not in direct contact with one another. We have modified our statements to reflect this fact. Our understanding is that the PLA is the best available method for detecting a direct or close interaction between two proteins in cell samples, and for quantitatively comparing the extent of such interactions and how they change (ie, over time as we did in our study).

2. Is the H53A GAPDH mutant expressed at comparable levels to WT?

Yes, in previous studies including HEK293 cells we have shown that its HA-tagged version is expressed to the same extent as HA-tagged wild type GAPDH. In any case, we have removed the H53A data from the revised manuscript, for reasons explained above.

3. The discussion states that GAPDH dissociates from mitochondria after it obtains heme. Was this shown?

It is based on our PLA data, that shows there is an increase in FLVCR1b-GAPDH interaction for 30 min after adding dALA/Fe that is kinetically correlated with the TC-GAPDH heme loading timeframe (takes 30 min), after which the FLVCR1b-GAPDH PLA interaction drops away. The connection to heme loading is inferred in this way.

4. Possible mechanisms for how TANGO2 might operate to support FLVCR1b transport should be discussed.

We now add in the discussion: "Conceivably, TANGO2 could help FLVCR1b acquire mitochondrial heme and/or facilitate heme passage through FLVCR1b." Its role will depend on its mitochondrial localization relative to FLVCR1b, which we now know is present in both the inner and outer mitochondrial membranes. This will help guide our work on TANGO2.

Reviewer #4 (Remarks to the Author):

I am typically very positive about studies coming from the Stuehr group, which has

made many contributions to the heme trafficking field in eukaryotes. Although in general I am a little less enthusiastic about this manuscript, there are some interesting observations here. Basically, the study involves a mitochondrial outer membrane protein FLVCR1b, which I refer to as FLVCR for my review. FLVCR has been implicated in heme transport for many years, yet the field does not seem to be unanimous. Here the authors use their cytoplasmic GAPDH reporters (of heme binding) to determine if FLVCR is needed for heme transport to it. They also carry out experiments to see if GAPDH and FLVCR interact. I like the approaches but I have some comments on their interpretations of results, and use of proper controls.

In general, I feel that they do not use the GAPDH H53A mutant enough as control. His53 is required by GAPDH to bind heme, and short of real biochemical reconstitution experiments with purified components, it may be the best way to prove if they really are on to something.

As we noted above, we decided to remove the H53A data from the revised manuscript, because we do not yet clearly know if H53A has defective interaction with FLVCR1b besides having a heme binding defect. This possibility will take significant further work to sort it out. If H53A is not solely a heme binding mutant it would complicate its use and interpretation.

My other general concern is the claim that they now have all the players in heme trafficking. How does heme get through the mitochondrial inner membrane, or the intermembrane space to FLVCR? Why do their results say "this completes an outline of the entire process" as they state in the Discussion?

We agree and now eliminate "entire". The intra-mitochondrial transport events are certainly important and we did not mean to ignore them. Historically, our group's interest in the intracellular heme trafficking pathway begins at the mitochondrial outer membrane, and we have not yet studied how it is transported inside this organelle. Now that our new data (included in the Revised) shows that FLVCR1b is also present in the inner membrane, it is conceivable it could help transport heme through both mitochondrial membranes.

Specific comments:

1. "Our results identify a transporter protein located in the outer mitochondrial membrane, Feline Leukemia Virus subgroup C Receptor 1b (FLVCR1b), as the sole conduit for mitochondrial heme export to GAPDH in the cells, operating through a mechanism that involves a direct FLVCR1b-GAPDH interaction for heme exchange. " Not sure if this claim is proven, particularly in view of the outer membrane association of FLVCR1b. As I state above, how does heme get from the ferrochelatase in side mitos to FLVCR1b?

We did not study heme trafficking within the mitochondria. As noted above, we have now found FLVCR1b is present in both the inner and outer mito membranes. This would be consistent with FLVCR1b enabling heme transport through both membranes. We plan to investigate this exciting possibility in our follow-up studies. In any case, we modified the statement to read: "Our results identify a transporter protein located in the outer mitochondrial membrane, Feline Leukemia Virus subgroup C Receptor 1b (FLVCR1b), as the conduit for mitochondrial heme export to GAPDH in the cells, operating through a mechanism that involves a FLVCR1b-GAPDH interaction for heme exchange. "

2. One aspect that is hard for me to understand, based on the authors statement: "The targeted siRNA treatment diminished cell FLVCR1b protein expression relative to a scrambled siRNA control by 70 +/- 9% (n = 5) (Fig. 1A and S1A) and this did not impact the cell heme level from increasing in response to the d-ALA/Fe addition (Table S1), consistent with a previous report. " So the FLVCR knockdown reduces expression by about 70% but the knockdown line/cell has the same levels of heme. Where is this heme, and how did it get there?

In the original Tolosano report on FLVCR1b, they also found that its knockdown had no significant effect on the cell's total heme level but did find it increased the mitochondrial heme level and diminished cytosolic heme level to some extent. So this implies the heme gets stuck inside the mitochondria. Our findings agree with theirs: FLVCR1b knockdown caused no change in the ability of the cell to make heme or in their total heme level.

Also, when ALA is added to induce heme levels ~ 3 fold, the knockdown prevents 90% of heme going to GAPDH. Don't know why 70 gets 90.

Generally, there is an inherent challenge of precisely quantifying the degree of change in Western band image intensities that makes it difficult to exactly relate the estimated change in protein expression derived in that way or to the degree of a change in some other biologic effect (in our case, heme loading into TC-GAPDH). In this context, 70% and 90% are reasonably close. A more interesting possibility is that the percentage differences reflect our new finding that FLVCR1b is in both the inner and outer mitochondrial membranes. If FLVCR1b is used to transport heme through both membranes, a 70% knockdown of FLVCR1b expression would have a greater % effect on the amount of heme that ultimately gets out to GAPDH (i.e., the effect of 70% knockdown on two consecutive membrane transfers; $0.3 \times 0.3 = 0.09$ to GAPDH).

3. Fig 1D, why is there such low heme (C14 labelled) in the "no ALA" cells and it appears tens of fold more in the +ALA siFLVCR line?

This sample has only background counts because these cells received no 14C-D-ALA. This is the negative control. The X axis titlenow indicates which cells did or did not receive 14C-dALA/Fe.

4. "Antibody pulldown experiments were done to independently assess the PLA results and they confirmed there is a temporal buildup in GAPDH-FLVCR1b association that peaked at 30 min after the d-ALA/Fe addition (Fig S3A and B). "

I know that the quantitations show some increase at the 30 min timepoint but looking at the Western itself, it seems underwhelming.

It is challenging to get good Western imaging data for FLVCR1b by these Ab co-IP's, due to the inherent low expression level of FLVCR1b in the samples. The Western images in Fig S9A provided only part of the data that generated the statistical result in panel B, and on balance the change in GAPDH band intensities at 30 min are consistent with about a 2.5x increase. Note that the Ab co-IP's were done to independently confirm the robust PLA results we obtained that show the change in GAPDH-FLVCR1b interaction. Both approaches indicate a buildup in the interaction of GAPDH with FLVCR1b at 30 min and a subsequent decay.

5. "In separate PLA-based experiments, we found that knockdown of FLVCR1b expression in the cells prevented GAPDH interaction with the mitochondrial surface protein HADHA and the increase in their interaction that otherwise occurred in response to d-ALA/Fe addition in the control cells (Fig. 2C and D). This confirmed that the changes in GAPDH interaction specifically involved its interaction with mitochondrial FLVCR1b."

Can this section be clarified? What is HADHA, why does it interact with GAPDH, why does siFLVCR reduce it? How does this show a direct interaction of GAPDH with FLVCR?

This has been clarified in the revised. As noted above in our reply to Rev. 1, our use of HADHA was not ideal and we have repeated the study using an established mitochondrial outer membrane protein (Hexokinase 1). These results are now included and show that knockdown of FLVCR1b diminishes GAPDH interaction with the mitochondrial surface protein HEX1, consistent with FLVCR1b being the reason that GAPDH interacts with the mitochondrial surface.

6. It appears that Fig2E somewhat contradicts the direct interaction (handoff of heme) from FLVCR to GAPDH: A His53 variant of GAPDH that does not bind heme still shows an increase in interaction with FLVCR upon ALA addition, albeit after 30 min (120min)

rather than at 30 min. (The pull-downs in Supp Fig 4 also suggest the interaction is independent of heme). Their conclusion is thus hard to rationalize: "This suggests that heme binding to GAPDH was coupled to the rise and fall in its interaction with FLVCR1b that occurred when cell mitochondrial heme biosynthesis is stimulated. "

As noted above in our previous replies, we don't know exactly why the H53A GAPDH interaction with FLVCR1b is muted and delayed. Although the difference in the time course of H53A interaction with FLVCR1b might be related to its lack of heme acquisition, an alternative explanation, as pointed out by the Reviewers, is that H53A may also have a defect in its ability to interact with FLVCR1b. Because this ambiguity makes the results with H53A difficult to interpret, we removed the H53A data from the revised manuscript.

7. Results in Fig 3 show transfer of heme to GAPDH and direct interaction with FLVCR. These experiments would be strengthened by using the GAPDH His53Ala variant as a control.

We agree that H53A could potentially be used to probe additional questions concerning how the heme transfer from isolated mitochondria to GAPDH is triggered or regulated. But we first will need to resolve if H53A possesses an additional defect in its FLVCR1b interaction (a possibility discussed in replies above). In any case, testing H53A is not essential for the main message of the results, which is that in a simple reconstitution system (i.e., one that only contains isolated mitochondria), adding GAPDH alone is enough to cause a significant and rapid export of heme from the mitochondria to the GAPDH (based on fluorescence decrease of TC-GAPDH). This clearly demonstrates that no other cell components needed to be added to trigger substantial heme transfer out of the isolated mitochondria to GAPDH in this system. This is important because it argues against the need for unidentified middleman proteins that might bind and transfer heme, or enable transfer of the heme between the mitochondria and GAPDH.

8. Seems like results in Fig1D and 4C are nearly identical, yet different experiments. Is this just coincidence or maybe a mix-up?

Thank you for catching this. It was a mix-up and the correct data set is now added in the revised Fig. 4C.

9. In Discussion the authors state: "Our current findings clarify its role by showing TANGO2 interacts with FLVCR1b and not with GAPDH and enables FLVCR1b export of mitochondrial heme to GAPDH. This role is consistent with knockout of TANGO2 causing an increase in the cell mitochondrial heme level 31, with TANGO2 displaying a

poor affinity toward binding heme 31, and with a recent study that cast doubt on its functioning in intracellular heme transport 36.

I am not sure how their data is consistent with both ref31 and ref 36, or how it clarifies the situation of TANGO2 as a heme chaperone. It would also help to know when they say it is in the mito, where is it (e.g. matrix, IMS, IM, OM)?

We agree that the critique of TANGO2 in Ref. 36 about its function in heme transport is less related to our work, so we no longer reference it in the revised. The submitochondrial localization of TANGO2 is unknown and we did not attempt to determine it because our study is primarily focused on FLVCR1b. Despite its location being unknown, our findings still advance understanding of TANGO2 by eliminating some possibilities: For example, our evidence indicating that cytosolic TANGO2 is not needed for FLVCR1b-mediated heme transfer to GAPDH and to downstream heme proteins argues against it acting as a cytosolic heme chaperone. Our finding that in mitochondrial isolates TANGO2 interacts with FLVCR1b but not with GAPDH argues against it functioning to give heme to GAPDH. Thus, our data provide new insight by narrowing down the possible functions of TANGO2 in the overall heme allocation process that we studied. Given this information and the reported poor heme binding affinity of TANGO2, we believe it reasonable and consistent with all the present data to speculate that TANGO2 may function to enable FLVCR1 transport of heme. This role would be consistent with a report showing that heme builds up in the mitochondria in the TANGO2 knockout and our finding that in TANGO2 knockdown cells the mitochondria do not release heme to GAPDH. Our current findings will spur more investigations focused on TANGO2 (including its mitochondrial localization and its impact on heme delivery to or transport through FLVCR1b).

NCOMMS-24-24607 Revision 2

Dear Editors:

Thank you for obtaining this second round of reviews of our manuscript “**Heme allocation in eukaryotic cells relies on mitochondrial heme export through FLVCR1b to cytosolic GAPDH**” by Drs. Jayaram *et al* and Stuehr. Our detailed replies to the second round of Reviews are below.

To recap what has happened so far: We responded to the first Reviewer’s comments on our original submitted manuscript by performing extensive additional studies, increasing manuscript clarity, and modifying our conclusions, as was described in our replies to the original reviews. This first round of revisions fully satisfied some of the Reviewers but also led one Reviewer to pose some follow-up suggestions and questions. In addition, I believe a Reviewer was recruited to comment on our replies to the first Reviewer 4 who was unavailable to participate in the second review. As is detailed below, we have responded to the remaining questions and suggestions by adding into the second revised manuscript new data, rearranging the presentation of some data, and adding in clarifications in the text. We thank the Editors and Reviewers for their effort in enabling the improvements.

Reviewer #1 (Remarks to the Author):

The authors made significant efforts to enhance the quality of the manuscript. We appreciate their work in improving the quality of the PLA images, refining the PLA controls, and validating the FLVCR1b-GAPDH and the FLVCR1b-TANGO2 interactions.

Author reply: Thank you for commending our efforts.

The Co-IP is fundamental to demonstrate the existence of an interaction between proteins because PLA permits detection of proximity between proteins (at distances < 40 nm). Thus, considering the relevance of this experiment, the Co-IP results (Fig. S9 and Fig. S14) should be placed among the main figures.

Author reply: We agree, to better highlight the FLVCR1b-GAPDH interaction we have moved the Co-IP Western data demonstrating their association (originally in Fig. S9) into the revised main Figure 2 as panels C and D.

Regarding Fig. S14, please note this is now S15 in the revised version. In this case we would like to keep the Co-IP Western results that show the TANGO2 interaction with FLVCR1b in Fig. S15, because the corresponding PLA data that indicates their association is also in Fig. S15.

Furthermore, the graphical representation of the Co-IP results should be improved to ensure more clarity. It should be clearly understood which protein has been immunoprecipitated and which one has been immunoblotted by simply observing the picture. For example, the graphical representation will be improved by adding a note on the top of Fig.S9A to show that the first 4 lanes on the left are the IgG isotype control and the 4 lanes on the right are the FLVCR1b IP. Same suggestion for Fig.S14. Please refer to published IP representation to improve the clarity of your picture.

Author reply: We have modified the graphical representations throughout as per this suggestion.

Concerning Fig.S9, why are IgG bands visible only in the IgG isotype controls and not where FLVCR1b has been immunoprecipitated? We cannot find information about the type of IgG in the methods. Which IgG has been used for control?

Author reply: We have used Goat IgG Isotype control (Thermo Scientific, #02-6202). It has been included in the methods section 'Immunoprecipitation of mitochondrial samples'. We understand your concern regarding the absence of IgG bands in the FLVCR1b immunoprecipitation samples. The IgG band may or may not be seen in the IP westerns depending on the specificity of antibodies used for IP to the target protein. If it does not cross-react with other proteins, including the antibody itself, one might not see significant IgG bands on the blot. However, upon a closer look on the blot imaged for FLVCR1b in Fig. 2C, we can still see a faint heavy chain corresponding to IgG band ~ 50 kDa.

Fig.S9E: this is another essential control of the specificity of the GAPDH-FLVCR1b interaction. Unfortunately, the resolution of the image is poor and GAPDH is difficult to detect. We recommend improving the picture. Also in this case, at a glance the image seems like a simple western blot. It would be better to represent the picture as a co-IP.

Author reply: We have replaced the original Western blot image with a new improved image in our revised Fig. S9. It now also includes IgG controls and added graphical representation to better indicate the picture is a co-IP.

Fig. S14: is it possible to improve the resolution of Fig. S14D?

Author reply: Yes, and we have increased the resolution of the mentioned figure. It is now Fig. S15D in the revised version.

Supplementary Figure S3. "Effect of FLVCR1b knockdown on the ability of FIAsh-TChGAPDH expressed in cells to bind exogenously-supplied versus endogenous heme". This is a very interesting experiment that should be emphasized in a main figure.

Author reply: Thank you for this suggestion. We have moved the results from the original Fig. S3 into the main Figure 1, panels E and F.

Minor comments:

- In the legend of Fig.S3 we read "vehicle plus heme (5 uM), or dALA + Fe (1mM and 100 μM) as indicated". The picture reporter FeCit whereas in the legend we can find only Fe. What does it mean "as indicated"? Maybe a clearer way to write it could be 1mM dALA + 100uM FeCit?*

Author reply: Thank you for pointing this out. We now correct this as 1 mM δALA + 100 uM FeCit.

- The authors used two different antibodies to detect FLVCR1a and FLVCR1b. Please indicate in the Methods section which antibody was used for each protein.*

Author reply: This information is now included in the methods section.

• *Fig. S1C: is it possible to improve the resolution of the picture? Maybe it is sufficient to modulate the intensity/saturation of the signal.*

Author reply: We have improved the resolution of the picture as suggested.

• *I cannot find information about the plasmid used to express Flvcr1b in the methods.*

Author reply: It has now been added into the methods section.

• *Please uniform text and figures: sometimes d-ALA/Fe sometimes d-ALA/FeCit.*

Author reply: It has now been made uniform as δ -ALA/Fe, after we fully define it in its first appearance in the text.

English revision is needed. Below please find some suggestions:

• *Lanes 70-71: "We also show that the protein TANGO2, likely through it interacting with FLVCR1b, is needed for the mitochondrial heme transfer to GAPDH". Change the underlying text with "through its interaction".*

• *Lane 85: "We then utilized an siRNA". Change an with a.*

• *Lane 85: "We then utilized an siRNA that can specifically knock down expression of FLVCR1b but otherwise was reported not to cause off-target effects 20 to determine how FLVCR1b knockdown would impact heme loading into TC-hGAPDH in response to our stimulating cell mitochondrial heme biosynthesis by providing the heme precursor molecules d-aminolevulinic acid (d-ALA) and ferric citrate, which in our hands typically causes the heme level in HEK293T cells and the level of heme bound in the TChGAPDH to increase by about 3-fold within 2 h". This sentence is too long. Please rephrase the sentence to improve clarity and readability.*

• *Lane 92-98: This sentence is too long. Please rephrase the sentence to improve clarity and readability.*

• *Lane 141: "A dual membrane location for FLVCR1b is consistent with it functioning to export heme from mitochondria to cytosolic GAPDH". Change it with its*

• *Lane 145: We next investigated if mitochondrial heme export to GAPDH involved it interacting with FLVCR1b by utilizing the Duolink proximity ligation assay (PLA). Correct the grammar. I suppose Duolink is a registered trademark so it should be written as Duolink ®.*

Author reply: Thank you for these suggestions, they are all incorporated.

Below are our Author replies to the new Rev.1's follow-up comments about our previous responses to the Rev. 4 comments from our first revision. Please note that the new Rev. 1 follow-up questions and our replies below are only the ones that new Rev 1 voiced concerns about. Out of our 11 total replies to Rev 4 from the last submission, the New Rev. 1 was fully satisfied with 7, and only those remaining below required our further response.

Key to understanding this section

Italic = the original Rev 4 comments

Bold italic = Our original replies to Rev 4

Normal font = New Rev 1 follow-up comments about our original replies to Rev 4 that New Rev 1 was not completely satisfied with.

Bold font = Author's new replies to the comments of new Rev 1

Original Rev 4 comment: "In separate PLA-based experiments, we found that knockdown of FLVCR1b expression in the cells prevented GAPDH interaction with the mitochondrial surface protein HADHA and the increase in their interaction that otherwise occurred in response to d-ALA/Fe addition in the control cells (Fig. 2C and D). This confirmed that the changes in GAPDH interaction specifically involved its interaction with mitochondrial FLVCR1b." Can this section be clarified? What is HADHA, why does it interact with GAPDH, why does siFLVCR reduce it? How does this show a direct interaction of GAPDH with FLVCR?

Original Author's reply: we repeated the study using an established mitochondrial outer membrane protein (Hexokinase 1). These new results are now included and show that knockdown of FLVCR1b diminishes GAPDH interaction with the mitochondrial surface protein HEX1, consistent with FLVCR1b being the reason that GAPDH interacts with the mitochondrial surface.

New Reviewer 1 comment: We are not convinced that this is the correct interpretation of the data. In our view, the experiment demonstrates that FLVCR1b is not required for the basal interaction between GAPDH and HK; however, it appears to prevent the heme-induced enhancement of this interaction. It seems that GAPDH can interact with multiple mitochondrial proteins, which may diminish the significance of the study's central finding—namely, the identification of a specific interaction with FLVCR1b that facilitates heme export from mitochondria.

Author's reply to new Rev 1 comment: We believe the central finding is that FLVCR1b is the major and possibly only conduit for mitochondrial heme transfer to GAPDH. This is important information because GAPDH is known to allocate mitochondrial heme to a wide range of cellular heme proteins. Our finding that GAPDH heme acquisition is linked to and requires an association between FLVCR1b and GAPDH on the mitochondrial surface is mechanistically important but derives from the primary finding. In any case, we performed the Hex1 experiment to test if FLVCR1b expression on the mito surface was important for enabling a functionally-important GAPDH-mito association (i.e., the GAPDH mito surface association that is tied to it receiving heme from the mitochondria). The results showed that the GAPDH-mito surface interaction that is tied to GAPDH acquiring mitochondrial heme no longer occurred when FLVCR1b expression was knocked down, and the GAPDH no longer acquired mitochondrial heme. This revealed that the specific GAPDH-FLVCR1b association at the mitochondrial surface that we saw increases during GAPDH heme acquisition is functionally important, because it enables the heme to transfer from the mitochondria to GAPDH.

Regarding the GAPDH-Hex1 association, it is reasonable to expect that when GAPDH associates with FLVCR1b on the mitochondrial surface it would be located close enough to other surface proteins like Hex1 to score positive for association within the precision

of the PLA assay. Whether such adjacent associations are functionally important is hard to say, but our current findings show that in the absence of FLVCR1 the GAPDH-Hex1 association is not functionally important. Thus, our observing some level of GAPDH-Hex1 association at the mitochondrial surface is reasonable and does not diminish from our finding that the specific FLVCR1b-GAPDH association is functionally critical for the heme transfer to GAPDH.

Original Rev 4 comment: In general, I feel that they do not use the GAPDH H53A mutant enough as control. His53 is required by GAPDH to bind heme, and short of real biochemical reconstitution experiments with purified components, it may be the best way to prove if they really are on to something.

Author reply to Rev 4: As we noted above, we decided to remove the H53A data from the revised manuscript, because we do not yet clearly know if H53A has defective interaction with FLVCR1b besides having a heme binding defect. This possibility will take significant further work to sort it out. If H53A is not solely a heme binding mutant it would complicate its use and interpretation.

New Rev 1 comment: We agree with the reviewer that the data concerning the H53A GAPDH mutant were unclear and difficult to interpret. As the authors did not address this issue, they have decided to remove these data from the revised version of the manuscript. Nevertheless, we believe that including this mutant—or others affecting either GAPDH-FLVCR1b interaction or heme binding—would serve as valuable controls and significantly strengthen the overall conclusions of the study.

Author reply to New Rev. 1 comment: We have re-included the original H53A GAPDH data as requested (Fig. S11) along with two new graphs that show its inability to bind heme in response to a heme titration or in response to increasing mitochondrial heme production in live cells (new panels A & B in Fig. S11). We believe this data is important to include for the readers. As shown by the original PLA and Co-IP Western data in Fig S11, the H53A mutant displays a delayed and diminished association with FLVCR1b on the mitochondrial surface after the cell mitochondrial heme production is stimulated. But as pointed out by the Reviewers we cannot conclude that its aberrant FLVCR1b association is actually due to it having defective heme binding, it is rather just a property associated with this behavior. Accordingly, we no longer suggest a causal relationship in the text and instead state: “Thus, a defective mitochondrial association and poor heme binding affinity of H53A HA-TC-GAPDH may likely explain why it was unable to receive mitochondrial heme in the cells.” Although we would like to determine if a connection ultimately exists between the heme binding ability of GAPDH and its level of mitochondrial FLVCR1b association in cells, we believe this question is secondary and that solving it is not required to support the main conclusions of our study.

Original Rev.4 comment: One aspect that is hard for me to understand, based on the authors statement: "The targeted siRNA treatment diminished cell FLVCR1b protein expression relative to a scrambled siRNA control by 70 +/- 9% (n = 5) (Fig. 1A and S1A) and this did not impact the cell heme level from increasing in response to the d-ALA/Fe addition (Table S1), consistent with

a previous report. " So the FLVCR knockdown reduces expression by about 70% but the knockdown line/cell has the same levels of heme. Where is this heme, and how did it get there?

Author reply to Rev 4: In the original Tolosano report on FLVCR1b, they also found that its knockdown had no significant effect on the cell's total heme level but did find it increased the mitochondrial heme level and diminished cytosolic heme level to some extent. So this implies the heme gets stuck inside the mitochondria. Our findings agree with theirs: FLVCR1b knockdown caused no change in the ability of the cell to make heme or in their total heme level.

New Rev 1 comment: The authors did not address this comment in their response. In the Tolosano work, it was shown that FLVCR1b knockdown did not alter total cellular heme levels but led to an increase in mitochondrial heme and a corresponding decrease in cytosolic heme. We believe the authors should replicate these findings in their experimental models to validate and strengthen their conclusions.

Author reply to New Rev 1 comment: We now include these measures in Fig S1. Our results match Tolosano et al's finding that FLVCR1b knockdown changes the mitochondrial vs cytosolic heme distribution in the predicted way.

Original Rev 4 comment: 7. Results in Fig 3 show transfer of heme to GAPDH and direct interaction with FLVCR. These experiments would be strengthened by using the GAPDH His53Ala variant as a control.

Authors reply to Rev 4: We agree that H53A could potentially be used to probe additional questions concerning how the heme transfer from isolated mitochondria to GAPDH is triggered or regulated. But we first will need to resolve if H53A possesses an additional defect in its FLVCR1b interaction (a possibility discussed in replies above). In any case, testing H53A is not essential for the main message of the results, which is that in a simple reconstitution system (i.e., one that only contains isolated mitochondria), adding GAPDH alone is enough to cause a significant and rapid export of heme from the mitochondria to the GAPDH (based on fluorescence decrease of TC-GAPDH). This clearly demonstrates that no other cell components needed to be added to trigger substantial heme transfer out of the isolated mitochondria to GAPDH in this system. This is important because it argues against the need for unidentified middleman proteins that might bind and transfer heme or enable transfer of the heme between the mitochondria and GAPDH.

New Rev 1 comment: See the comment at the beginning.

Authors reply to New Rev 1 comment: Please see our related reply above.

In general, I feel that they do not use the GAPDH H53A mutant enough as control. His53 is required by GAPDH to bind heme, and short of real biochemical reconstitution experiments with purified components, it may be the best way to prove if they really are on to something.

As we noted above, we decided to remove the H53A data from the revised manuscript, because we do not yet clearly know if H53A has defective interaction with FLVCR1b besides having a heme binding defect. This possibility will take significant further work to sort it out. If H53A is not solely a heme binding mutant it would complicate its use and interpretation.

We agree with the reviewer that the data concerning the H53A GAPDH mutant were unclear and difficult to interpret. As the authors did not address this issue, they have decided to remove these data from the revised version of the manuscript.

Nevertheless, we believe that including this mutant—or others affecting either GAPDH-FLVCR1b interaction or heme binding—would serve as valuable controls and significantly strengthen the overall conclusions of the study.

My other general concern is the claim that they now have all the players in heme trafficking. How does heme get through the mitochondrial inner membrane, or the intermembrane space to FLVCR? Why do their results say "this completes an outline of the entire process" as they state in the Discussion?

We agree and now eliminate “entire”. The intra-mitochondrial transport events are certainly important and we did not mean to ignore them. Historically, our group’s interest in the intracellular heme trafficking pathway begins at the mitochondrial outer membrane, and we have not yet studied how it is transported inside this organelle. Now that our new data (included in the Revised) shows that FLVCR1b is also present in the inner membrane, it is conceivable it could help transport heme through both mitochondrial membranes.

The authors’ reply is exhaustive.

Specific comments:

1. *"Our results identify a transporter protein located in the outer mitochondrial membrane, Feline Leukemia Virus subgroup C Receptor 1b (FLVCR1b), as the sole conduit for mitochondrial heme export to GAPDH in the cells, operating through a mechanism that involves a direct FLVCR1b-GAPDH interaction for heme exchange. "* Not sure if this claim is proven, particularly in view of the outer membrane association of FLVCR1b. As I state above, how does heme get from the ferrochelatase in side mitos to FLVCR1b?

We did not study heme trafficking within the mitochondria. As noted above, we have now found FLVCR1b is present in both the inner and outer mito membranes. This would be consistent with FLVCR1b enabling heme transport through both membranes. We plan to investigate this exciting possibility in our follow-up studies. In any case, we modified the statement to read: "Our results identify a transporter protein located in the outer mitochondrial membrane, Feline Leukemia Virus subgroup C Receptor 1b (FLVCR1b), as the conduit for mitochondrial heme export to GAPDH in the cells,

operating through a mechanism that involves a FLVCR1b-GAPDH interaction for heme exchange. "

The authors' reply is exhaustive.

2. One aspect that is hard for me to understand, based on the authors statement: "The targeted siRNA treatment diminished cell FLVCR1b protein expression relative to a scrambled siRNA control by 70 +/- 9% (n = 5) (Fig. 1A and S1A) and this did not impact the cell heme level from increasing in response to the d-ALA/Fe addition (Table S1), consistent with a previous report. " So the FLVCR knockdown reduces expression by about 70% but the knockdown line/cell has the same levels of heme. Where is this heme, and how did it get there?

In the original Tolosano report on FLVCR1b, they also found that its knockdown had no significant effect on the cell's total heme level but did find it increased the mitochondrial heme level and diminished cytosolic heme level to some extent. So this implies the heme gets stuck inside the mitochondria. Our findings agree with theirs: FLVCR1b knockdown caused no change in the ability of the cell to make heme or in their total heme level.

The authors did not address this comment in their response.

In the Tolosano work, it was shown that FLVCR1b knockdown did not alter total cellular heme levels but led to an increase in mitochondrial heme and a corresponding decrease in cytosolic heme.

We believe the authors should replicate these findings in their experimental models to validate and strengthen their conclusions.

Also, when ALA is added to induce heme levels ~ 3 fold, the knockdown prevents 90% of heme going to GAPDH. Don't know why 70 gets 90.

Generally, there is an inherent challenge of precisely quantifying the degree of change in Western band image intensities that makes it difficult to exactly relate the estimated change in protein expression derived in that way or to the degree of a change in some other biologic effect (in our case, heme loading into TC-GAPDH). In this context, 70% and 90% are reasonably close. A more interesting possibility is that the percentage differences reflect our new finding that FLVCR1b is in both the inner and outer mitochondrial membranes. If FLVCR1b is used to transport heme through both membranes, a 70% knockdown of FLVCR1b expression would have a greater % effect on the amount of heme that ultimately gets out to GAPDH (i.e., the effect of 70% knockdown on two consecutive membrane transfers; $0.3 \times 0.3 = 0.09$ to GAPDH).

The authors' reply is exhaustive. I do not expect a perfect correspondence between reduction of protein level and reduction of substrate transport.

3. Fig 1D, why is there such low heme (C14 labelled) in the "no ALA" cells and it appears tens of fold more in the +ALA siFLVCR line?

This sample has only background counts because these cells received no 14C-D-ALA. This is the negative control. The X axis title now indicates which cells did or did not receive 14C-dALA/Fe.

The authors' reply is exhaustive.

4. "Antibody pulldown experiments were done to independently assess the PLA results and they confirmed there is a temporal buildup in GAPDH-FLVCR1b association that peaked at 30 min after the d-ALA/Fe addition (Fig S3A and B). "

I know that the quantitations show some increase at the 30 min timepoint but looking at the Western itself, it seems underwhelming.

It is challenging to get good Western imaging data for FLVCR1b by these Ab co-IP's, due to the inherent low expression level of FLVCR1b in the samples. The Western images in Fig S9A provided only part of the data that generated the statistical result in panel B, and on balance the change in GAPDH band intensities at 30 min are consistent with about a 2.5x increase. Note that the Ab co-IP's were done to independently confirm the robust PLA results we obtained that show the change in GAPDH-FLVCR1b interaction. Both approaches indicate a buildup in the interaction of GAPDH with FLVCR1b at 30 min and a subsequent decay.

The authors' reply is exhaustive.

5. "In separate PLA-based experiments, we found that knockdown of FLVCR1b expression in the cells prevented GAPDH interaction with the mitochondrial surface protein HADHA and the increase in their interaction that otherwise occurred in response to d-ALA/Fe addition in the control cells (Fig. 2C and D). This confirmed that the changes in GAPDH interaction specifically involved its interaction with mitochondrial FLVCR1b."

Can this section be clarified? What is HADHA, why does it interact with GAPDH, why does siFLVCR reduce it? How does this show a direct interaction of GAPDH with FLVCR?

This has been clarified in the revised. As noted above in our reply to Rev. 1, our use of HADHA was not ideal and we have repeated the study using an established mitochondrial outer membrane protein (Hexokinase 1). These results are now included and show that knockdown of FLVCR1b diminishes GAPDH interaction with the mitochondrial surface protein HEX1, consistent with FLVCR1b being the reason that GAPDH interacts with the mitochondrial surface.

We are not convinced that this is the correct interpretation of the data. In our view, the experiment demonstrates that FLVCR1b is not required for the basal interaction between GAPDH and HK; however, it appears to prevent the heme-induced enhancement of this interaction.

It seems that GAPDH can interact with multiple mitochondrial proteins, which may diminish the significance of the study's central finding—namely, the identification of a specific interaction with FLVCR1b that facilitates heme export from mitochondria.

6. *It appears that Fig2E somewhat contradicts the direct interaction (handoff of heme) from FLVCR to GAPDH: A His53 variant of GAPDH that does not bind heme still shows an increase in interaction with FLVCR upon ALA addition, albeit after 30 min (120min) rather than at 30 min. (The pulldowns in Supp Fig 4 also suggest the interaction is independent of heme). Their conclusion is thus hard to rationalize: "This suggests that heme binding to GAPDH was coupled to the rise and fall in its interaction with FLVCR1b*

that occurred when cell mitochondrial heme biosynthesis is stimulated. "

As noted above in our previous replies, we don't know exactly why the H53A GAPDH interaction with FLVCR1b is muted and delayed. Although the difference in the time course of H53A interaction with FLVCR1b might be related to its lack of heme acquisition, an alternative explanation, as pointed out by the Reviewers, is that H53A may also have a defect in its ability to interact with FLVCR1b. Because this ambiguity makes the results with H53A difficult to interpret, we removed the H53A data from the revised manuscript.

7. Results in Fig 3 show transfer of heme to GAPDH and direct interaction with FLVCR. These experiments would be strengthened by using the GAPDH His53Ala variant as a control.

We agree that H53A could potentially be used to probe additional questions concerning how the heme transfer from isolated mitochondria to GAPDH is triggered or regulated. But we first will need to resolve if H53A possesses an additional defect in its FLVCR1b interaction (a possibility discussed in replies above). In any case, testing H53A is not essential for the main message of the results, which is that in a simple reconstitution system (i.e., one that only contains isolated mitochondria), adding GAPDH alone is enough to cause a significant and rapid export of heme from the mitochondria to the GAPDH (based on fluorescence decrease of TC-GAPDH). This clearly demonstrates that no other cell components needed to be added to trigger substantial heme transfer out of the isolated mitochondria to GAPDH in this system. This is important because it argues against the need for unidentified middleman proteins that might bind and transfer heme, or enable transfer of the heme between the mitochondria and GAPDH.

See the comment at the beginning.

8. Seems like results in Fig1D and 4C are nearly identical, yet different experiments. Is this just coincidence or maybe a mix-up?

Thank you for catching this. It was a mix-up and the correct data set is now added in the revised Fig. 4C.

Done.

9. In Discussion the authors state: "Our current findings clarify its role by showing TANGO2 interacts with FLVCR1b and not with GAPDH and enables FLVCR1b export of mitochondrial heme to GAPDH. This role is consistent with knockout of TANGO2 causing an increase in the cell mitochondrial heme level 31, with TANGO2 displaying a poor affinity toward binding heme 31, and with a recent study that cast doubt on its functioning in intracellular heme transport 36.

I am not sure how their data is consistent with both ref31 and ref 36, or how it clarifies the situation of TANGO2 as a heme chaperone. It would also help to know when they say it is in the mito, where is it (e.g. matrix, IMS, IM, OM)?

We agree that the critique of TANGO2 in Ref. 36 about its function in heme transport is less related to our work, so we no longer reference it in the revised. The submitochondrial localization of TANGO2 is unknown and we did not attempt to determine it because our study is primarily focused on FLVCR1b. Despite its location being unknown, our findings still advance understanding of TANGO2 by eliminating

some possibilities: For example, our evidence indicating that cytosolic TANGO2 is not needed for FLVCR1b-mediated heme transfer to GAPDH and to downstream heme proteins argues against it acting as a cytosolic heme chaperone. Our finding that in mitochondrial isolates TANGO2 interacts with FLVCR1b but not with GAPDH argues against it functioning to give heme to GAPDH. Thus, our data provide new insight by narrowing down the possible functions of TANGO2 in the overall heme allocation process that we studied. Given this information and the reported poor heme binding affinity of TANGO2, we believe it reasonable and consistent with all the present data to speculate that TANGO2 may function to enable FLVCR1 transport of heme. This role would be consistent with a report showing that heme builds up in the mitochondria in the TANGO2 knockout and our finding that in TANGO2 knockdown cells the mitochondria do not release heme to GAPDH. Our current findings will spur more investigations focused on TANGO2 (including its mitochondrial localization and its impact on heme delivery to or transport through FLVCR1b).

The reply is acceptable even if the role of TANGO2 would necessitate additional work.